# Both microRNA-455-5p and -3p repress hypoxia-inducible factor-2α expression and coordinately regulate cartilage homeostasis

Yoshiaki Ito[1,2], Tokio Matsuzaki[3], Fumiaki Ayabe[3], Sho Mokuda[3], Ryota Kurimoto[1], Takahide Matsushima [1], Yusuke Tabata[1], Maiko Inotsume[1], Hiroki Tsutsumi[1], Lin Liu[1], Masahiro Shinohara[1], Yoko Tanaka[1], Ryo Nakamichi[3], Keiichiro Nishida[4], Martin K. Lotz [3] & Hiroshi Asahara [1,3✉]

Osteoarthritis (OA), the most common aging-related joint disease, is caused by an imbalance between extracellular matrix synthesis and degradation. Here, we discover that both strands of microRNA-455 (miR-455), -5p and -3p, are up-regulated by Sox9, an essential transcription factor for cartilage differentiation and function. Both miR-455-5p and -3p are highly expressed in human chondrocytes from normal articular cartilage and in mouse primary chondrocytes. We generate miR-455 knockout mice, and find that cartilage degeneration mimicking OA and elevated expression of cartilage degeneration-related genes are observed at 6-months-old. Using a cell-based miRNA target screening system, we identify hypoxia-inducible factor-2α (HIF-2α), a catabolic factor for cartilage homeostasis, as a direct target of both miR-455-5p and -3p. In addition, overexpression of both miR-455-5p and -3p protect cartilage degeneration in a mouse OA model, demonstrating their potential therapeutic value. Furthermore, knockdown of HIF-2α in 6-month-old miR-455 knockout cartilage rescues the elevated expression of cartilage degeneration-related genes. These data demonstrate that both strands of a miRNA target the same gene to regulate articular cartilage homeostasis.

[1] Department of Systems Biomedicine, Tokyo Medical and Dental University (TMDU), Bunkyo-ku, Tokyo, Japan. [2] Research Core, Tokyo Medical and Dental University (TMDU), Bunkyo-ku, Tokyo, Japan. [3] Department of Molecular Medicine, The Scripps Research Institute, La Jolla, CA, USA. [4] Department of Orthopaedic Surgery, Science of Functional Recovery and Reconstruction, Okayama University Graduate School of Medicine and Dentistry and Pharmaceutical Sciences, Okayama city, Okayama, Japan. ✉email: asahara@scripps.edu

Osteoarthritis (OA) is characterized by the progressive loss of articular cartilage. It can affect all joint tissues, leading to joint dysfunction and pain, and systemic and organismal changes that reduce the quality of life, initiate or aggravate co-morbidities, and reduce life expectancy[1]. The current treatment for OA is generally limited to invasive joint replacement and symptomatic treatment; therefore, there is a critical need to identify new disease mechanisms that could be used to develop more targeted therapies. In OA, cartilage cellularity in OA is reduced by chondrocyte death, and the remaining chondrocytes are activated by cytokines and growth factors, leading to the initiation of catabolic processes and abnormal differentiation/functions that degrade the extracellular matrix (ECM)[2–5]. Molecular mechanisms that govern articular chondrocyte differentiation during the development and maintenance of articular cartilage are being characterized, and this has the potential to lead to new therapeutic interventions.

miRNAs are a class of small noncoding RNAs that play critical roles in biological processes as negative regulators of gene expression by promoting messenger RNA (mRNA) degradation and/or repressing translation of protein-coding mRNAs[6–14]. miRNAs play a critical role in both cartilage development and homeostasis and are involved in OA pathogenesis. The miR-140 shows a cartilage-specific expression pattern, and its expression is induced by Sox9[15]. It is strongly expressed in normal mature articular cartilage and its expression level is significantly reduced in OA cartilage[16–18]. Furthermore, miR-140 knockout mice show the phenotype of early onset of OA-like changes[19], suggesting that miR-140 acts to maintain cartilage homeostasis. Other miRNAs that are expressed in cartilage have been identified and shown to regulate differentiation, cell survival, production of inflammatory cytokines, and matrix-degrading enzymes[20].

Usually, only one strand of Dicer-dependent miRNA duplex is incorporated into the RNA-induced silencing complex (RISC) to form functional and mature miRNA complexes. This selection is thought to be based on the thermodynamic parameters of that strand[21–23]. However, recent reports note that in exceptional cases, two distinct miRNAs can be generated, although their functional relevance is not fully understood[14,24–26].

In this study, we demonstrate that the expressions of both miR-455-5p and -3p from the same precursor miR-455 (pre-miR-455) are upregulated by overexpression of Sox9. Recently, it has been reported that aged miR-455-3p knockout mice show significant degeneration of knee cartilage, indicating that miR-455-3p is a critical cartilage homeostasis regulator[27]. However, the mechanisms of miR-455 functions in cartilage homeostasis remain largely unknown. In this study, we generated miR-455 knockout mice and confirmed an age-related OA-like phenotype similar to the previous report. We also investigated the targets of miR-455s by the reporter library system, which is a cell-based screening system for targets of post-transcriptional regulators. This system identified Hif-2α, a major catabolic factor for cartilage, as a target of both miR-455-5p and -3p. The knockdown of HIF-2α in 6-month-old miR-455 knockout cartilage rescues the elevated expression of cartilage degeneration-related genes. Moreover, to examine the therapeutic effect of miR-455s, we performed overexpression of miR-455s in OA-model mice. The results strongly implicate miR-455-5p and -3p in supporting articular cartilage homeostasis by targeting Hif-2α.

## Results

### miR-455s expression is regulated by Sox9 in chondrocytes.

We previously reported that chondrocyte-specific expression of miR-140 is tightly regulated by transcription factor Sox9, the chondrocytes master regulator, via the Sox9-binding enhancer region near the miR-140 gene[15]. To examine whether other miRNAs could also be regulated by Sox9 in chondrocytes, we comprehensively explored differentially expressed miRNAs in adenoviral-mediated Sox9 overexpressing murine chondrocytes by microarray. The microarray identified five miRNAs whose expression was increased, and 13 miRNAs whose expression was repressed by overexpression of Sox9 (Fig. 1a, Supplementary Data 1). The altered expression of these miRNAs by Sox9 overexpression was validated by quantitative RT-PCR (qPCR). The expression of Sox9 and Col2a1, a major target of Sox9, was promoted by adenoviral-mediated Sox9 overexpression at MOI 50 in chondrocytes (Supplementary Fig. 1). Under these conditions, 14 (5 upregulated and 9 repressed) out of the 18 identified miRNAs showed similar expression changes to the microarray data (Fig. 1b).

Among the five upregulated miRNAs, miR-455, as well as miR-140, was upregulated by Sox9 overexpression in both 5p and 3p strands (Fig. 1a, b). To investigate whether the expression of miR-455 is directly regulated by Sox9, we performed chromatin immunoprecipitation (ChIP) analysis for Sox9 in mouse chondrocytes. miR-455 is located in intron 10 of Col27a1. Intronic miRNAs are believed to be processed from the introns of their hosting transcription units and hence share common regulatory mechanisms and expression patterns with the host gene[28–30]. Since cartilage is a tissue conserved in mammals, we considered that the Sox9-binding sequence exists in the genomic region conserved among mammals. To search the transcriptional regulating region of Col27a1, we first investigated the conservation of the genomic sequence near the transcriptional start site of Col27a1 (−20 K to +20 K). The genomic sequence was compared with that of human, chimpanzee, rhesus, cow, and dog by VISTA-point[31]. We identified 16 conserved regions and created 19 primer sets for the ChIP (Supplementary Fig. 2a and Supplementary Table 1). ChIP using the anti-Sox9 antibody in mouse primary chondrocytes revealed that Sox9 binds a conserved region of Col27a1 intron 3 (primer set no. 16) (Supplementary Fig. 2b). Previous reports indicated that Sox9 homodimers bind to enhancer regions that contain inverted Sox9-binding sites separated by 3–4 bp and represent a palindromic motif for chondrocyte-specific gene regulation[32–34]. This region contained a palindromic sequence similar to the Sox9-binding consensus sequence separated by 4 bp (Supplementary Fig. 2c). These data suggest that miR-455 and the host gene Col27a1 are directly regulated by Sox9 in chondrocytes.

Next, we measured miR-455-5p and -3p expression in human articular chondrocytes and human mesenchymal stem cells (MSCs) by qPCR. Both miR-455-5p and -3p were significantly more highly expressed in human chondrocytes (ten different preparations from ten different donors) compared with MSCs (seven different preparations from three different donors) (Supplementary Fig. 3). In addition, we measured miR-455s expression during chondrogenesis of mouse MSCs in pellet cultures. The expression of both miR-455-5p and -3p was increased in 1-week differentiated cells compared to 0 week (Supplementary Fig. 4). The peaks of early chondrogenesis genes such as Sox9, Col2a1, and Acan peaked at 2 weeks of differentiation, whereas miR-455 had an early peak at the first week (Supplementary Fig. 4), indicating that miR-455 is expressed at the early stage of chondrogenesis.

We also investigated the ratio of miRNA-5p and -3p expression of ten miRNAs in mouse primary chondrocytes. These ten were selected to include five conserved miRNAs whose expression was increased more than 1.2-fold by infection with Sox9-expressing adenovirus at MOI = 5, and the top five conserved miRNAs that were highly expressed in both chondrocytes infected with Sox9 or LacZ-expressing adenovirus at

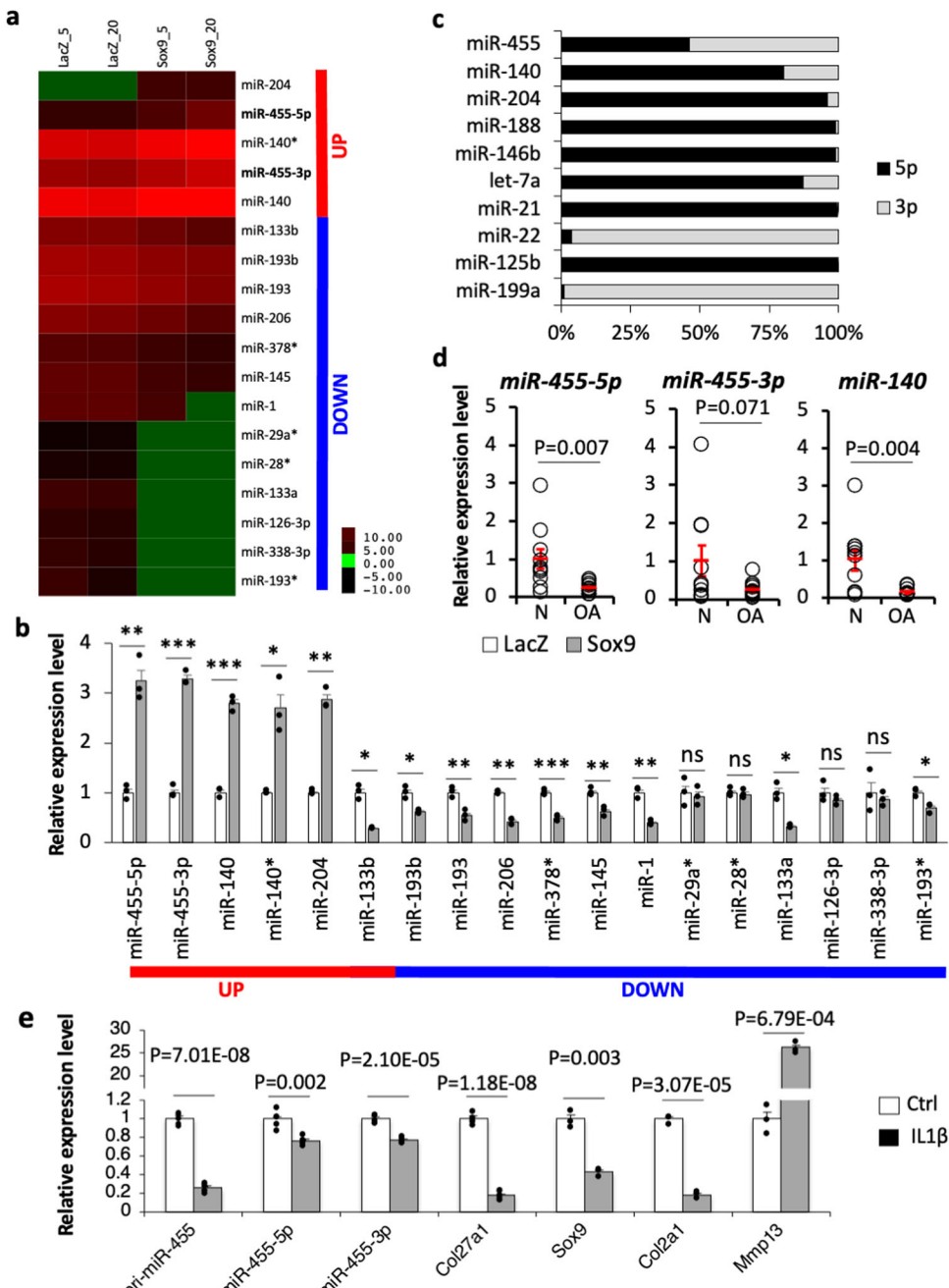

**Fig. 1 Expressions of miR-455-5p and -3p in chondrocytes. a** Heatmap of miRNA microarray. The microarray data of differentially expressed miRNAs (Signal value: > 1.5-fold and < 0.6-fold change in MOI = 5, and > threefold and < 0.6-fold change in MOI = 20) whose expression changed by adenovirally expressed Sox9 are shown. LacZ-expressing chondrocytes were used as a negative control. **b** Relative expression of miRNAs in Sox9- or LacZ-expressing adenovirus-infected chondrocytes. The experiment was performed once with three biological replicates ($n = 3$). Data are represented as the mean ± SEM. ns not significant, *$P < 0.05$, **$P < 0.01$, ***$P < 0.001$. Two-tailed Student's $t$ test. **c** Relative expression of both strands of miRNAs in mouse primary chondrocytes. The experiment was performed independently twice with two biological replicates ($n = 4$). **d** Relative expression of miR-455s and miR-140 in OA ($n = 11$, biologically independent samples) or normal articular cartilage (N; $n = 10$, biologically independent samples). Data are represented as the mean ± SEM. Two-tailed Student's $t$ test. **e** Relative expression of pri-miR-455, *Col27a1*, mature miR-455s, *Sox9*, *Col2a1* and *Mmp13* in mouse primary chondrocytes with or without IL-1β stimuli (IL-1β or Ctrl). For miR-455s and *Col27a1*, the experiment was performed independently twice with two or three biological replicates ($n = 5$). For other cartilage-related genes, the experiment was performed once with three biological replicates ($n = 3$). Data are represented as the mean ± SEM. Two-tailed Student's $t$ test. Source data are provided as a Source Data file.

MOI = 5. As shown in Fig. 1c, unlike the other miRNAs, miR-455 produced almost the same amount of both the 5p and 3p strands in chondrocytes. The high level of expression of both strands of miR-455 in human chondrocytes, along with its regulation by Sox9, suggest that both miR-455s are functional regulators of chondrocytes.

**miR-455s expression is suppressed in OA articular cartilage and IL-1β stimulation**. To examine changes in expression of the miR-455s in OA articular cartilage, qPCR of miR-455s together with miR-140 was performed on 21 samples prepared from human knee articular cartilage (10 normal and 11 OA). As expected, the expression of miR-140 was significantly decreased

in OA cartilage (Fig. 1d). miR-455-5p expression in articular cartilage from donors with OA (obtained from individuals 57–89 years old; Mankin score 5–10) was also significantly lower than in normal cartilage (obtained from individuals 21–55 years old; Mankin score 0–2) (Fig. 1d). miR-455-3p expression in OA samples was also reduced, although the difference was not statistically significant (Fig. 1d).

The cytokine IL-1β plays a critical role in OA, and IL-1β stimulation of chondrocytes causes gene expression patterns similar to those in OA cartilage[35]. Thus, we analyzed the effect of IL-1β stimulation on miRNA-455s using qPCR. IL-1β significantly suppressed primary miR-455 (pri-miR-455), mature miR-455-5p/3p, Col27a1, Sox9, and Col2a1, whereas expression of the cartilage degenerative marker Mmp13 was significantly increased (Fig. 1e). Consistent with the above analysis of OA samples, these results indicate that catabolic signals, i.e., IL-1β stimulation in chondrocytes, lead to the reduction of both miR-455s along with the host gene expression.

**OA-like pathology is observed in miR-455 knockout mouse knee joints.** In order to investigate the in vivo function of miR-455 in OA, we generated miR-455 knockout mice using CRISPR/Cas9 genome engineering. Guide RNA targeting miR-455-5p sequences and Cas9 mRNA were microinjected in BDF1 mouse embryos and generated mice with a 22 bp deletion that included the miR-455-5p mature sequences (Fig. 2a and Supplementary Fig. 5). Neither miR-455-5p nor -3p was expressed in primary chondrocytes generated from miR-455[−/−] mice (Supplementary Fig. 6). The miR-455 gene is located in intron 10 of the Col27a1 gene, but the expression of Col27a1 did not change significantly between wild-type and miR-455 knockout chondrocytes (Supplementary Fig. 6). The miR-455[−/−] mice were born normally, and the bodyweight of postneonatal day 0 (P0) and 4-week-old mice was similar to wild-type mice (Supplementary Fig. 7). Because chondrocytes play a critical role in skeletal development, we investigated skeletal formation in miR-455[−/−] mice by performing skeletal prep with neonatal miR-455[−/−] and wild-type mice. Skeletal development was normal in miR-455 knockout mice (Supplementary Fig. 8). We also performed micro-CT scanning in 8-week-old male miR-455[−/−] mice (Supplementary Fig. 9). There were no significant changes in bone morphometry and microarchitecture of the femur and skull in miR-455[−/−] mice. Bone volume and mineral density in the femur were not significantly changed in miR-455[−/−] mice. In addition, gross tissue size analysis and histological analysis for major tissues (thyroid, pancreas, spleen, heart, lung, kidney, liver, and testis) were performed in 8-week-old miR-455 knockout mice, but no difference was observed between wild-type and knockout mice (Supplementary Fig. 10 and Supplementary Table 2).

Chondrocytes play a critical role not only in skeletal and cartilage formation but also in articular cartilage homeostasis. We found reduced miR-455s expression in human OA cartilage, suggesting that miR-455s may regulate OA pathology. To examine the potential role of miR-455s in cartilage homeostasis, we investigated cartilage degeneration in the knee joints in miR-455[−/−] mice. At 2-month-old (2-mo), no significant change was observed between miR-455 knockout and wild-type mouse knee joints (Fig. 2b, c). However, at 6 mo, miR-455[−/−] knee joints showed cartilage disruption, and the OARSI score (osteoarthritis cartilage histopathology assessment system, developed by the Osteoarthritis Research Society International)[36] was significantly higher in miR-455 knockout mice than in wild-type mice (Fig. 2b, c). Furthermore, we performed qPCR analysis for cartilage marker and degenerative genes of knee cartilage in wild-type and miR-455 knockout mice. In 2-month-old mice, no differences in

expression of these genes were observed between wild-type and knockout mice; however, the expression of Mmp3, Mmp13, and Adamts5, major cartilage degenerative genes were significantly increased in 6-month-old miR-455 knockout mice (Fig. 2d). These data indicate that miR-455 is critical for cartilage homeostasis.

**The reporter library system identified EPAS1 as a target of miR-455s.** To elucidate the molecular mechanisms of miR-455 functions, we screened miR-455 targets using a reporter library system. This system is a cell-based target screening system for post-transcriptional regulators that uses a reporter library composed of 4891 full-length human cDNAs, each of which was integrated into the 3′UTR of a luciferase gene (Fig. 3a)[37]. Screening is conducted by luciferase assays on the reporter library with or without an expression vector for the pri-miRNA of interest. This system evaluates the full-length sequence of the targets including the 5′ untranslated region (5′UTR) and the coding region at the translational level. We identified 12 reporters with this system and validated them by 96-well scale luciferase assays. As a result, we identified four reported (CALR, CPEB1, RAF1, and SMAD2)[7,38–40] and five previously unreported (APH1A, EPAS1, FBXL2, and ZBTB20) target candidates of miR-455 (Fig. 3b). These were validated in mouse primary chondrocytes, where overexpression of miR-455-5p and/or -3p mimics reduced their expression (Fig. 3c). Also, luciferase assays using fragmented reporters indicated that miR-455s regulates EPAS1, APH1A, and ZBTB20 mainly via their 3′UTR, and FKBPL and FBXL2 were regulated via their coding region (Fig. 3d).

Among these targets, we focused on EPAS1, which encodes HIF-2α. HIF-2α is known as a catabolic transcription factor for cartilage homeostasis[41,42]. EPAS1 has the seed sequence in the 3′ UTR for both miR-455s. To test whether EPAS1 is directly regulated by miR-455s, we performed a reporter assay using the luciferase vector containing the EPAS1 3′UTR sequence with point mutations in the putative miR-455s-binding sequences (Fig. 4a). As expected, the luciferase activity of the wild-type EPAS1 3′UTR reporter decreased in the presence of a pri-miR-455 expression vector (Fig. 4b). However, this was rescued in reporter vectors carrying point mutations in both miR-455-5p and -3p target sites, or in either miR-455-5p or miR-455-3p target sites (Fig. 4b). We next investigated Hif-2α expression in miR-455 knockout mouse cartilage. The ratio of Hif-2α expressed chondrocytes in 2- and 6-mo miR-455 knockout knee cartilage was significantly increased compared to wild-type chondrocytes (Fig. 4c, d). In addition, the increased expression of Hif-2α in 6-mo miR-455 knockout cartilage was rescued by overexpression of miR-455-5p and -3p mimics (Fig. 4e, f). These data reveal that both miR-455-5p and -3p directly regulate EPAS1 expression, and suggest that both miR-455s have an anti-inflammatory function and protect against cartilage destruction in OA[41–43].

**Introduction of both miR-455-5p and -3p inhibits cartilage degeneration.** To investigate the potential therapeutic effect of miR-455s, we used the well-established surgical destabilization of the medial meniscus (DMM) model of OA injected with miR-455s mimics (Fig. 5a). We used the transfection method for the introduction of miRNAs, which is safer than using viral vectors and has lower hurdles for medical applications. We first investigated whether the introduction of miRNA mimics had an impact on articular cartilage and the phenotype of the surrounding tissue in the absence of DMM surgery. The cartilage degeneration score and the subchondral bone score of the negative control mimic- or miR-455s-introduced group did not differ from those of the sham group (Supplementary Fig. 11a–c). Regarding the surrounding

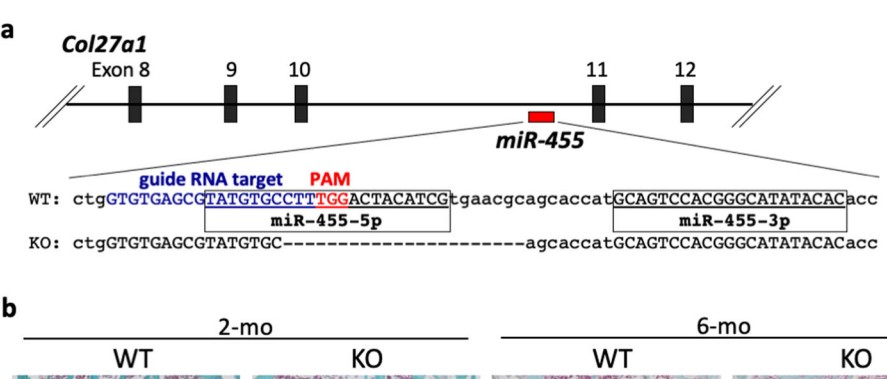

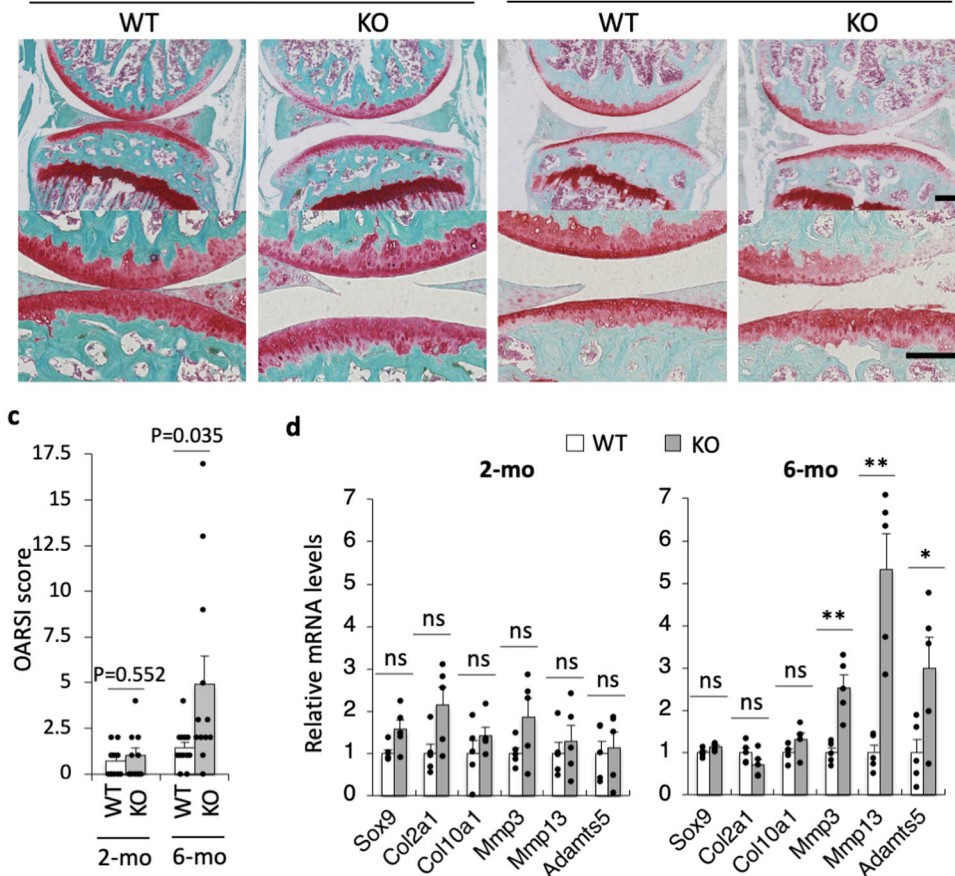

**Fig. 2 miR-455 knockout mice show an OA-like phenotype. a** Genetic deletion in miR-455-deficient mice generated by CRISPR/Cas9. **b** Representative image of Safranin-O staining of wild-type (WT) and miR-455 knockout (KO) mice (2, 6-month-old). Staining was repeated at least three times with similar results. Scale bars show 200 μm. **c** The OARSI scores of 2 (n = 10, biologically independent samples) or 6-month-old (n = 12, biologically independent samples) wild-type (WT) and miR-455 knockout (KO) mice (2, 6-month-old). Data are represented as the mean ± SEM. Two-tailed Student's *t* test. **d** Relative mRNA levels of cartilage markers and cartilage degeneration-related genes in knee cartilage of wild-type (WT) and miR-455 knockout (KO) mice (2, 6-month-old). Data are represented as the mean ± SEM, n = 5 biologically independent samples. ns: not significant, \*P < 0.05, \*\*P < 0.01. Two-tailed Student's *t* test. Source data are provided as a Source Data file.

muscles (quadriceps femoris), no difference was observed between the sham group and the miRNA-transfected group (Supplementary Fig. 12). These data indicate that injection of the miRNA mimic does not affect the phenotype of the knee joint. Injection of both miR-455-5p and -3p mimics into DMM-treated knee joints significantly inhibited cartilage destruction compared to injection of control mimics (Fig. 5b, c). Moreover, Hif-2α, a target of miR-455-5p and -3p, and its downstream catabolic enzyme, Mmp13, were downregulated in DMM-treated knee cartilage in the miR-455s-injected group (Fig. 5d, e). These results reveal a therapeutic effect of miR-455-5p and -3p for treating cartilage degeneration in OA, possibly by repressing Hif-2α expression.

**Knockdown of *Epas1* in miR-455 knockout knee cartilage rescues the abnormally increased expression of cartilage degeneration-related genes.** To investigate whether the phenotypes associated with cartilage degeneration in miR-455 knockout mice are due to the increased expression of *Epas1* in the cartilage, we performed *Epas1* knockdown by siRNA in 6-month-old KO mouse knee cartilage (Fig. 6a). The expression of *Epas1* in knee cartilage of 6-month-old wild-type and KO mice was significantly reduced by knockdown of *Epas1* (Fig. 6b). Moreover, the expression of *Mmp3* and *Mmp13*, which have been reported to be downstream target genes of Hif-2α[41,42], was significantly decreased by *Epas1* knockdown (Fig. 6b). Similarly, the expression of *Nos2*, which is also considered to be a downstream target

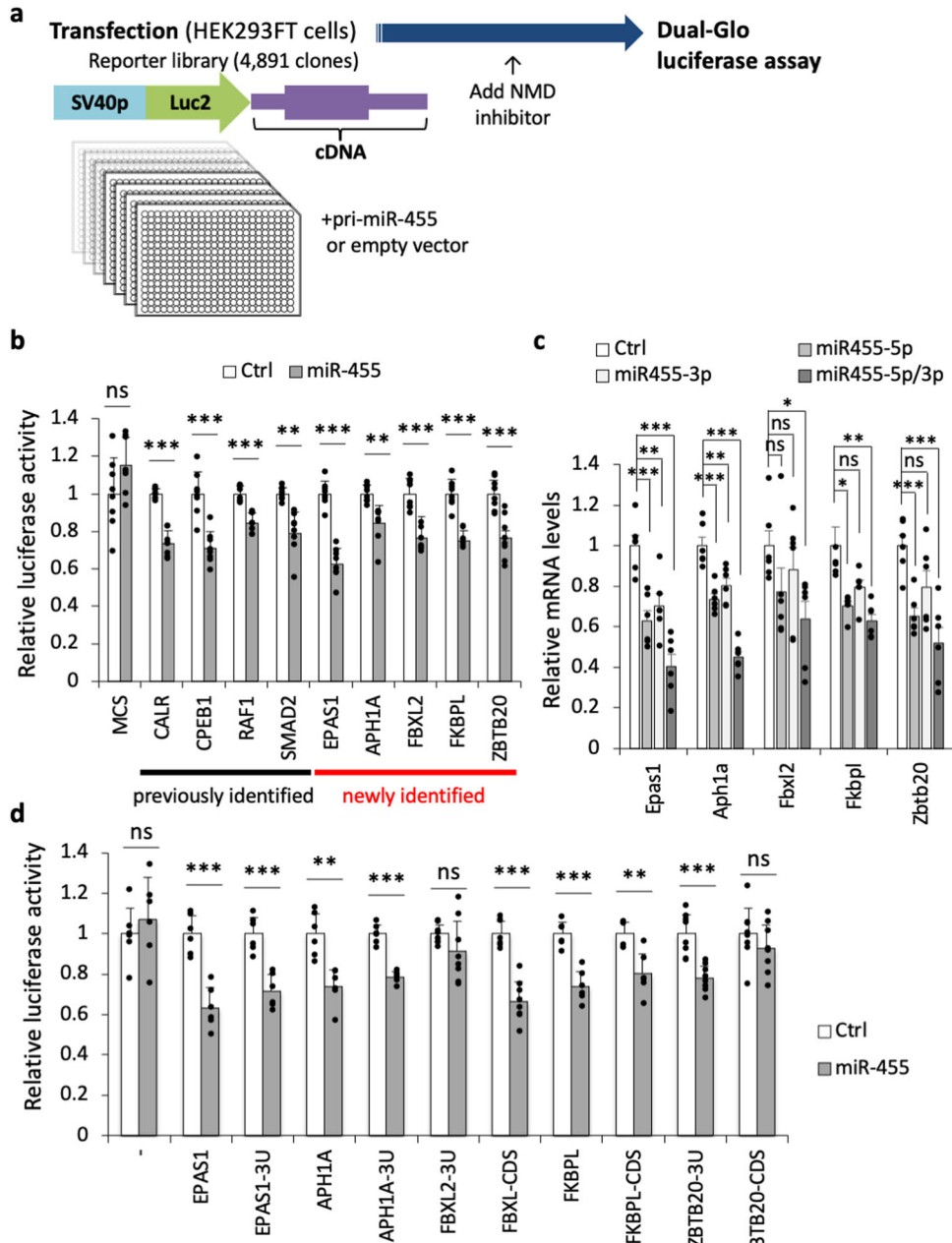

**Fig. 3 Identifying the targets of miR-455s using a reporter library system. a** A schematic model of the reporter library system for screening for miR-455 targets. **b** Luciferase assays of various reporters in 293FT cells transfected with pcDNA-miR-455 (miR-455) or the empty vector (Ctrl). MCS, pLuc2-KAP-MCS (empty reporter). The assays were performed independently four times in duplicate (n = 8). Data are represented as the mean ± SD. ns not significant, **P < 0.01, ***P < 0.001. Two-tailed Student's t test. **c** Relative mRNA levels of identified target candidates in mouse primary chondrocytes transfected with the negative control miRNA mimic (Ctrl), miR-455-5p mimic (miR-455-5p), miR-455-3p mimic (miR-455-3p) or both miR-455-5p and -3p mimics (miR-455-5p/3p). The experiment performed independently three times with two biological replicates (n = 6). Data are represented as the mean ± SEM. ns not significant, *P < 0.05, **P < 0.01, ***P < 0.001. Two-tailed Dunnett's test. **d** Luciferase assays of whole and fragmented reporters in 293FT cells transfected with pcDNA-miR-455 (miR-455) or empty vector (Ctrl). Error bars show SD. The assays were performed independently three or four times in duplicate (n = 6–8). Data are represented as the mean ± SD. ns not significant, **P < 0.01, ***P < 0.001. Two-tailed Student's t test. -: empty reporter, 3U: 3′UTR, CDS: coding region. Source data are provided as a Source Data file.

gene of Hif-2α[42], and a cartilage degeneration marker gene, *Adamts5*, were decreased by *Epas1* knockdown in miR-455 KO knee cartilage (Fig. 6b), although no significant differences were observed (P = 0.0573, and P = 0.0569, respectively). These data indicate that the phenotype of cartilage degeneration in miR-455 knockout mice is due, at least in part, to the increased expression of *Epas1* in the cartilage.

## Discussion

In this study, we provide evidence that both miR-455-5p and -3p are expressed in chondrocytes by being directly regulated by Sox9, and that expression of both miR-455s is reduced in OA cartilage and in response to IL-1β stimulation. The aged miR-455s knockout mice show an OA-like phenotype, indicating the significance of miR-455s in cartilage homeostasis. In addition, we

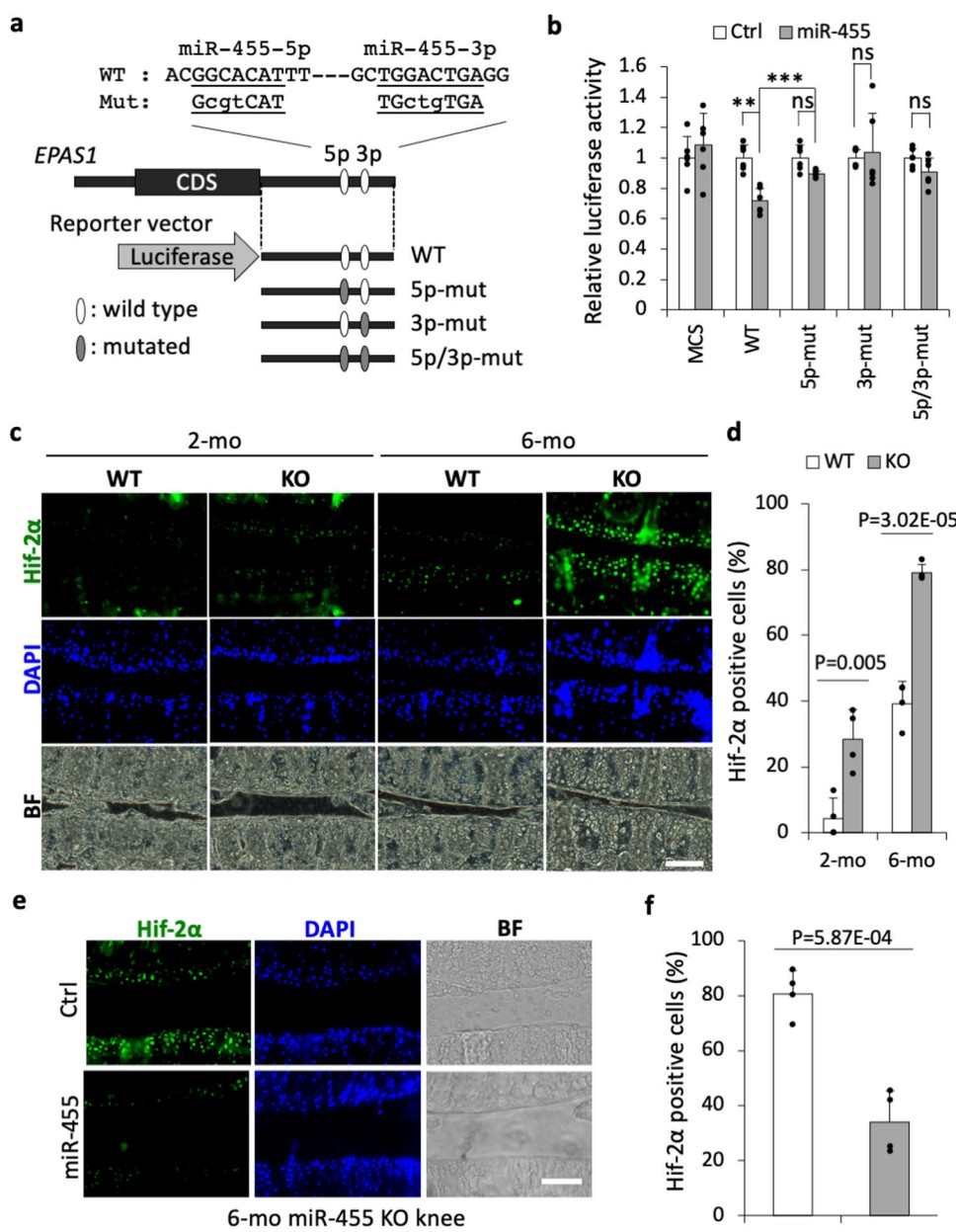

**Fig. 4 EPAS1 is a direct target of both miR-455-5p and -3p. a** The four *EPAS1* reporters tested in the luciferase assay. mut sequences with a point mutation, WT wild-type sequences. **b** Luciferase assays of various *EPAS1* reporters in 293FT cells transfected with pcDNA-miR-455 (miR-455) or the empty vector (Ctrl). MCS, pLuc2-KAP-MCS (empty reporter). ns not significant. The assays were performed independently three times in duplicate ($n=$ 6). Data are represented as the mean ± SD. ns not significant, *$P < 0.05$, **$P < 0.01$, ***$P < 0.001$. Two-tailed Tukey–Kramer test. **c** Representative image of Hif-2α, DAPI staining and bright-field (BF) of wild-type (WT) and miR-455 knockout (KO) articular cartilage (2, 6-month-old). Staining was repeated at least twice with similar results. Scale bars show 100 μm. **d** The percentage of Hif-2α positive cells in wild-type (WT) and miR-455 knockout (KO) articular cartilage (2, 6-month-old). Data are represented as the mean ± SD, $n = 4$, biologically independent samples. Two-tailed Student's *t* test. **e** Representative image of Hif-2α and DAPI staining and bright-field view (BF) of 6-mo miR-455 knockout articular cartilage transfected with control mimic (Ctrl) or miR-455-5p and -3p mimics (miR-455). Staining was repeated twice with similar results. Scale bars show 100 μm. **f** The percentage of Hif-2α positive cells in 6-mo miR-455 knockout articular cartilage transfected with control mimic (Ctrl) or miR-455-5p and -3p mimics (miR-455). Data are represented as the mean ± SD, $n = 4$. Two-tailed Student's *t* test. Source data are provided as a Source Data file.

demonstrated that HIF-2α, a central transactivator that targets several crucial catabolic genes[41–43], is regulated by miR-455s. Furthermore, we showed that injection of miR-455s mimics in the knee joint inhibits DMM-induced cartilage destruction.

We demonstrated that the expression of miR-455s is reduced in OA cartilage and is directly promoted by Sox9. The expression of miR-140 is also regulated by Sox9 in chondrocytes, and miR-140 and miR-455s show similar expression in OA cartilage

compared to normal cartilage[16,44]. It has been demonstrated that miR-455-3p expression is reduced in OA cartilage[27], which is consistent with our data. On the other hand, there is a report that both miR-140 and miR-455-3p show increased expression in OA cartilage from the hip[7]. The difference in these results may reflect the stage of the OA samples, and the location and type of cartilage that was collected from OA joints, which is supported by published data[45–48].

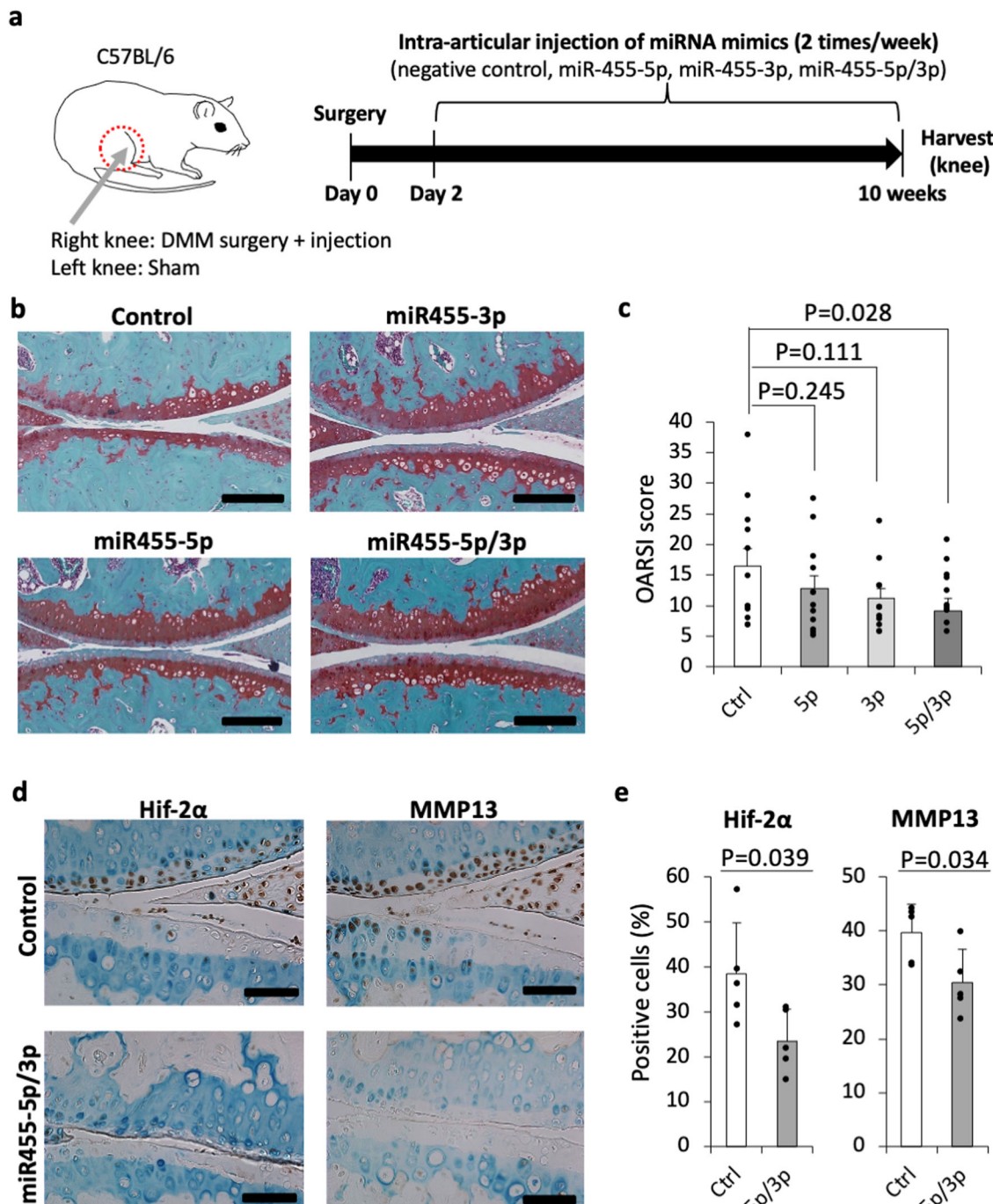

**Fig. 5 Potential therapeutic effect of miR-455s in OA. a** Schedule of treatment using transfection of miR-455s mimic in a surgically induced OA model. The mouse figure is a partial modification of the figure created by Sho Mokuda in the previous paper[65]. **b** Representative image of Safranin-O staining. Staining was repeated at least twice with similar results. Scale bars show 200 μm. **c** OARSI scores. Data are represented as the mean ± SEM, $n = 12$, biologically independent samples. One-tailed Dunnett's test. Ctrl: control, 5p: miR-455-5p, 3p: miR-455-3p, 5p/3p: miR-455-5p and -3p. **d** Representative image of Hif-2α or MMP13 staining. Staining was repeated at least twice with similar results. Scale bars show 100 μm. **e** Percentages of Hif-2α and MMP13-positive cells. Data are represented as the mean ± SD, $n = 5$, biologically independent samples. Two-tailed Student's *t* test. Ctrl: control, 5p/3p: miR-455-5p and -3p. Source data are provided as a Source Data file.

This study identified both miR-140 and miR-455s as miRNA targets strongly upregulated by Sox9. Sox9 is a master transcription factor for chondrogenesis expressed in developing cartilage[49–51]. Sox9 remains expressed in articular chondrocytes and regulates ECM homeostasis by regulating the expression of major cartilage-specific matrix proteins, including type II and type XI collagen and aggrecan[52], although the expression of Sox9 is reduced in OA cartilage[53,54]. The repression of catabolic factors, Adamts5 and Hif-2α, via targeting by miR-140 and miR-455s may be a key role, together with ECM gene regulation, for Sox9 in cartilage homeostasis.

We identified *EPAS1*, *APH1A*, *FBXL2*, *FKBPL*, and *ZBTB20* as target genes of miR-455 by using a reporter library system. Although we focused on *EPAS1* alone in this study, it is

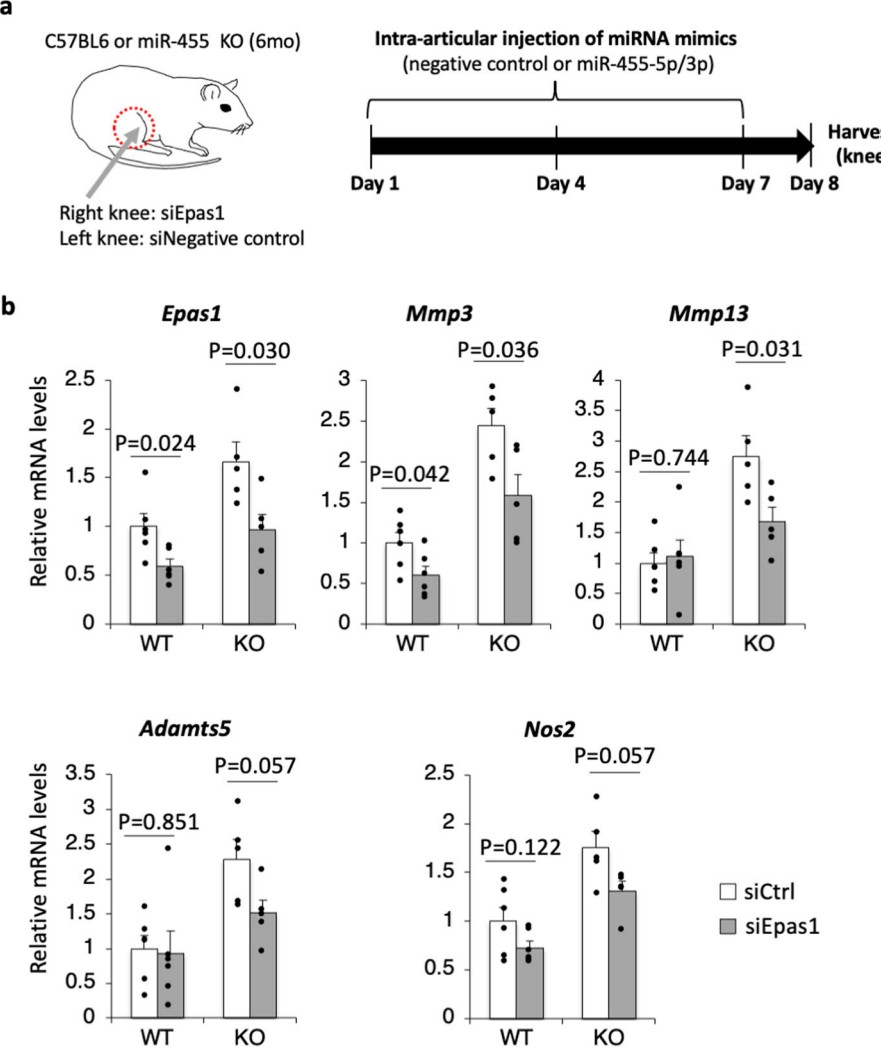

**Fig. 6 Knockdown of *Epas1* in miR-455 KO knee cartilage rescues the abnormally increased expression of cartilage degeneration-related genes.**
**a** Schedule of treatment using transfection of siRNA for *Epas1* in knee cartilage of 6-month-old miR-455 knockout mice. The mouse figure is a partial modification of the figure created by Sho Mokuda in the previous paper[65]. **b** Relative expression levels of *Epas1*, *Mmp3*, *Mmp13*, *Adamts5*, and *Nos2* in siRNA-transfected knee cartilage of 6-month-old wild-type (WT; n = 6, biologically independent samples) or miR-455 knockout mice (KO; n = 5, biologically independent samples). Data are represented as the mean ± SEM. Two-tailed Student's *t* test. siCtrl: negative control siRNA, siEpas1: siRNA for *Epas1*. Source data are provided as a Source Data file.

conceivable that some of the other genes may be important target genes mediating the inhibitory effect of miR-455 on cartilage degeneration. For example, Zbtb20 has been reported to be important in the terminal differentiation of hypertrophic chondrocytes by repressing Sox9[55]. It is possible that Zbtb20 is involved in cartilage homeostasis as well as development by Sox9 repression, and that miR-455 may contribute to cartilage homeostasis also by repressing Zbtb20.

HIF-2α is a homolog of HIF-1α and a member of the basic helix–loop–helix/PAS transcription factor family[56]. It has been implicated in osteoarthritis pathogenesis. HIF-2α has an abnormal expression pattern in OA cartilage and regulates hypertrophic differentiation of OA chondrocytes[41]. *Epas1* $^{+/-}$ mice show significant resistance to cartilage destruction and overexpression of Hif-2α enhances cartilage degradation[41,42]. Hif-2α directly promotes the expression of catabolic genes such as *Mmp3*, *Mmp13*, *Col10a1*, and *Nos2*[41,42]. These data indicate that HIF-2α causes cartilage destruction by transactivating catabolic genes. In this study, we demonstrated that miR-455s directly regulates HIF-2α expression. In addition, we also find that miR-

455 knockout mice at 6 month showed destruction of knee cartilage and enhanced expression of Hif-2α and its target genes *Mmp3* and *Mmp13* in knee cartilage. Moreover, the introduction of miR-455-5p and -3p mimics prevented DMM-induced cartilage destruction and reduced Hif-2α expression. Furthermore, knockdown of *Epas1* in articular cartilage of miR-455 knockout mice rescues the abnormally increased expression of cartilage degeneration-related genes, *Mmp3* and *Mmp13*. These data demonstrate that miR-455s regulate cartilage homeostasis likely by repressing Hif-2α expression and may represent a therapeutic target for OA.

The miR-455 knockout mice showed an OA-like phenotype, which is consistent with a recent report of 5-mo miR-455-3p deletion mice[27]. The mice in that report were generated by partial deletion of the miR-455-3p sequence, which is expected to inhibit the formation of the pre-miR-455 structure, and thus inhibit expression of both miR-455-5p and -3p. This report also indicated that miR-455-3p directly targets *PAK2*, which they showed was an inhibitor of TGF-β/Smad signaling that plays a key role in maintaining cartilage homeostasis[27]. They also demonstrated that

the expression of *PAK2* is upregulated in OA cartilage[27]. *PAK2* knockdown as well as overexpression of miR-455-3p in human OA chondrocytes correlated with increased expression of cartilage-specific genes *SOX9*, *COL2A1*, and *ACAN*, and decreased expression of cartilage degeneration-related genes *RUNX2*, *COL10A1*, and *MMP13*[27]. Thus, together with our data, we propose that PAK2 and HIF-2α are both likely to be critical genes for cartilage degeneration. Targeting these two genes by miR-455 might contribute more effectively to maintain cartilage homeostasis. We propose that the suppression of different genes that promote cartilage degeneration by two miRNAs generated from one pre-miRNA more effectively controls cartilage homeostasis.

Many miRNAs are predominantly produced from only one strand of the miRNA duplex[57–61]. The selection of the strand which is incorporated into RISC appears to derive from the lower thermodynamic parameters of that strand[21–23]. Thus, our finding that both miR-455-5p and miR-455-3p are equivalently expressed and functional is unusual. However, there are some reports from cancer studies that both arms can be expressed and functional. For example, the miR-145-5p and -3p coordinately target *MTDH*, whose high expression reduces lung squamous cell carcinoma patients[25]. Also, the miR-199a-5p and -3p are expressed at high levels in SWI/SNF chromatin complex gene *Brm*-deficient tumor cells but only marginally in *Brm*-expressing tumor cells, and both miR-199a-5p and miR-199a-3p directly repress *Brm* via its 3′ UTR[26]. The biological significance of generating two miRNAs from a pre-miRNA might be that two arms more reliably and stably targeting the same gene more effectively suppresses the expression of the target gene.

Some studies have indicated a therapeutic effect for overexpression of miRNA in in vivo OA models. For example, the overexpression of miR-142-3p by lentivirus-mediated gene transfer in an OA model significantly alleviated OA progression and inhibited NF-kB signaling and proinflammatory cytokines[62]. Intra-articular injection of miR-222-overexpressing lentivirus into the murine knee joints in the DMM OA model attenuated cartilage degradation and repressed MMP13 expression[63]. Also, intra-articularly injected miR-140 alleviated OA progression in a trauma-induced early-stage OA rat model[64]. In this study, we suggest that there may be an additive effect of miR-455-5p and -3p in the treatment of experimental OA. Both miR-455-5p and -3p coordinately regulate the expression of HIF-2α, a central transactivator of catabolic factors[41,42]. Although HIF-2α, encoded by the *EPAS1* gene, is a potential therapeutic target for OA since *EPAS1* is an important gene for development, *Epas1*[−/−] mice are embryonic lethal, and *Epas1*[+/−] mice show dwarfism[41]. Therefore, indirect suppression of *EPAS1* by mir-455s may be safer than direct siRNA knockdown of *EPAS1* as a treatment for OA. Also, injection of both miR-455-5p and -3p into intra-articular cartilage was more effective in alleviating experimental OA compared to injection of miR-455-5p or -3p alone, further supporting their additive activity. These data also indicate that using multiple miRNAs in OA treatment can be more functional. In this regard, we have also demonstrated the additive effect of a miRNA and the host mRNA in an OA model[65]. These data indicate that combining multiple miRNA injections may lead to more effective treatments for OA.

## Methods

**Study approval**. Human and animal studies received ethical approval by the Scripps Human Subjects Committee at The Scripps Research Institute and Institutional Animal Care and Use Committee at The Scripps Research Institute and Tokyo Medical and Dental University. Human tissues and cells were obtained with approval by the Scripps Human Subjects Committee at The Scripps Research Institute. The study design and conduct complied with all relevant regulations regarding the use of human study participants and was conducted in accordance with the criteria set by the Declaration of Helsinki. All study donors provided written informed consent. All animal studies were performed according to protocols approved by Institutional Animal Care and Use Committee at The Scripps Research Institute and Tokyo Medical and Dental University. All mice were freely allowed access to food, water, and activity. Mice were maintained under a 12 h dark–light cycle and constant temperature (20–26 °C) and humidity maintenance (40–60%).

**Cell culture**. Mouse chondrocytes were prepared from embryonic day 16.5 (E16.5) or P0 ribs by digestion with collagenase. The primary chondrocytes and HEK293FT cells (Thermo Fisher; R70007) were maintained in DMEM containing 10% FBS and 1% penicillin/streptomycin at 37 °C in a humidified atmosphere with 5% $CO_2$. For stimulation with IL-1β, chondrocytes were cultured in a six-well plate and treated with IL-1β (10 ng/mL) for 24 h. Human chondrocytes were isolated from ten normal donors (mean ± SD age 40.2 ± 11.0 years; female four, male six) by digestion with collagenase. The primary chondrocytes were cultured in DMEM containing 10% FBS and 1% penicillin/streptomycin at 37 °C in a humidified atmosphere with 5% $CO_2$[66]. Experiments with chondrocytes were performed in passages 0–2. Human bone marrow-derived MSCs were purchased from Lonza (PT-2501), and some MSC preparations have previously been isolated from iliac crest bone marrow obtained from normal adult donors[67]. The MSCs were cultured in MSC basal medium and subcultured by treatment with Accutase (Innovative Cell Technologies)[67]. Experiments with MSCs were performed in passages 2–5.

**Chondrogenesis of mouse MSCs in pellet cultures**. Mouse MSCs were purchased from cyagen (MUBMX-01001) and cultured in a complete growth medium (cyagen, MUXMX-90011). Mouse MSCs were digested in 0.5% trypsin-EDTA (Wako), and trypsin was inactivated with 2× volume of expansion medium. Dissociated cells were centrifuged at 150×g for 5 min, and the supernatant was aspirated. Subsequently, cells were washed in incomplete chondrogenic induction medium (Lonza, PT-3003), and resuspended at $5 \times 10^5$ cells/mL in complete chondrogenic medium: incomplete chondrogenic induction medium and 10 ng/mL rhTGF-β3 (Lonza, PT-4124). In total, 500 μL of the above cell mixture was dispensed into 15-mL conical tubes and centrifuged at 200 ×g for 5 min. Pellets were cultured at 37 °C in 5% $CO_2$ for 21 days with medium exchanged every 3 days. Each pellet was homogenized in TRI Reagent (Molecular Research Center) using a BioMasher (Nippi). Subsequent steps were performed using the manufacturer's protocol.

**MicroRNA microarray**. Total RNA was extracted from cultured cells with ISO-GEN (Nippon gene) according to the manufacturer's protocol. The miRNA expression profiles were analyzed by Agilent standard protocol. Briefly, 100 ng of total RNAs were used per sample. Mature miRNAs were selectively labeled and hybridized on Mouse miRNA Microarray Release 15.0 (Agilent Technology) using microRNA Labeling and a Hybridization kit (Agilent Technology).

**Human tissues**. Human articular cartilage specimens were obtained from the knee joints of ten normal donors (mean ± SD age 40.2 ± 11.0 years; female four, male six) and from 11 OA grade IV donors (71.7 ± 11.4 years; female six, male five) who recently deceased. All samples were examined by Safranin-O staining and graded according to a modified Mankin scale[68], with a score of < 2 points being normal and a score of > 5 representing OA. The total RNA was isolated from fresh-frozen cartilage by homogenizing the tissue in a freezer mill (Spex) and extracting the homogenate in TRIzol (Invitrogen).

**Chromatin immunoprecipitation analysis**. Mouse primary chondrocytes were crosslinked with 1% formaldehyde for 10 min and quenched with 0.125 M glycine for 5 min. Cells were washed and lysed with lysis buffer (10 mM Tris-HCl, pH 7.4; 10 mM NaCl; 5 mM $MgCl_2$; 0.2% NP-40) and resuspended with 100 μl of glycerol buffer (10 mM Tris-HCl, pH 7.4; 0.1 mM EDTA; 5 mM $MgAc_2$; 25% glycerol) and 100 μl of reaction buffer (100 mM Tris-HCl, pH 7.4; 50 mM KCl; 8 mM $MgCl_2$; 2 mM $CaCl_2$). Chromatin was sheared by Micrococcal nuclease, stopped reaction by EGTA, and diluted with 1000 μl of IP buffer (25 mM Tris-HCl, pH 8.0; 150 mM NaCl; 2 mM EDTA; 1% Triton X-100; 0.1% SDS). The chromatin solution was incubated with 5 μg of anti-SOX9 antibody (Millipore, ab5535)- or normal rabbit IgG (Wako, 148-09551)-conjugated Dynabeads Protein A (Invitrogen) at 4 °C for 4 h. Beads were washed with low-salt wash buffer (20 mM Tris-HCl, pH 8.0; 150 mM NaCl; 2 mM EDTA; 1% Triton X-100; 0.1% SDS) four times and high-salt wash buffer (20 mM Tris-HCl, pH 8.0; 500 mM NaCl; 2 mM EDTA; 1% Triton X-100; 0.1% SDS), and immune-complexes were eluted from beads with ChIP elution buffer (50 mM Tris-HCl, pH 8.0; 10 mM EDTA; 1% SDS) at 65 °C. Eluates were additionally incubated at 65 °C to reverse cross-linking and then incubated with proteinase K at 55 °C. DNA was purified by QIAquick PCR purification kit (Qiagen). Aliquots and whole-cell extracts (serving as input samples) were analyzed by qPCR amplification. Primer sequences are shown in Supplementary Table 1.

**Generation of miR-455 knockout mice using CRISPR/Cas9 system**. The guide RNA (gRNA) containing target sequence of miR-455 (GTG TGA GCG TAT GTG

CCT T) and hCas9 mRNA was synthesized in vitro using mMESSAGE mMA-CHINE T7 Kit (Life Technologies) and purified MegaClear Kit (Life Technologies) according to the manufacturer's instruction. A mixture containing 125 ng/µl of gRNA and 250 ng/µl of hCas9 mRNA was microinjected into the cytoplasm of a one-cell stage BDF1 embryo. Genotyping PCR was performed using the following primers: miR-455 genotyping F, 5′-GCT TCC TTC CAC AGG TCG CG-3′; miR-455 genotyping R, 5′-CTC TGT GGT GGT GGC TCC AG-3′. PCR products were treated with EXO-SAP-IT (Affymetrix) and then used for direct sequencing with genotyping primer.

**OA-model mice**. C57BL/6J mice were used for this study. In the treatment study, OA was surgically induced by DMM in the right knee joints of 3-mo mice[69]. The left knees were subjected to sham surgery. The target mimic-conjugated hydrogel was created as described previously[65]. In brief, 1.5 µl of miRNA mimic (50 µM), 1.5 µl of completion buffer, and 3 µl of invivofectamine 3.0 reagent (Thermo) were mixed, followed by incubation at 50 °C for 30 min. Then, 15 µl PBS, 7.5 µl Atelo-Gene QG (KOKEN), and 7.5 µl dilution buffer were added. In all, 30 µl of the mixture was injected into the intra-articular space of the knee. The mice (n = 48) were divided into four groups. Group 1 (control group) was treated with a control miRNA mimic (Pre-miR, Ambion). Group 2 was treated with mouse miR-455-3p (Ambion). Group 3 was treated with mouse miR-455-5p (Ambion). Group 4 was treated with miR-455-3p and miR-455-5p. Two days after DMM, the treatment of mimic with an injection to the right knee started two times a week for 10 weeks and the mice were killed at 10 weeks after surgery. The entire knee joints were fixed in 10% zinc-buffered formalin for 2 days and decalcified in TBD-2 for 24 h. To investigate whether the mimic reached the cartilage, we created two additional groups that were treated with only mimic injection two times a week for 10 weeks. One was treated with control miRNA mimic, and the other with miR-455-5p and -3p (n = 3, each). The efficacy of the miR-455 mimic was examined by qPCR from RNA extraction from cartilage 3 days after the final injection (Supplementary Fig. 13). To test whether the injection of the miRNA mimic affects the phenotype, we created three additional groups that were treated only with mimic injection two times a week for 10 weeks. Group 1 was treated with control miRNA mimic (n = 6), Group 2 with miR-455-5p and -3p (n = 8), and group 3 was subjected to sham surgery (n = 6). The entire knee joints were fixed in 10% formalin for a day and decalcified in formic acid/sodium citrate solution for 24 h. The samples were embedded and stained with Safranin-O or hematoxylin and eosin.

**Histological analysis**. For H&E staining, tissues from 8-week-old male mice were fixed in 4% paraformaldehyde/0.1 M phosphate buffer, and embedded in paraffin. Sections of 4 µm in thickness were stained with hematoxylin (131-09665, FUJI-FILM Wako Pure Chemical Corp.) and eosin (051-06515, FUJIFILM Wako Pure Chemical Corp.). Mouse knee joints were harvested, fixed, decalcified, embedded, and stained with Safranin-O. The histological OA scores for medial femoral condyle, the medial tibial plateau, and summed scores of femur and tibia were evaluated using the Osteoarthritis Research Society International (OARSI) cartilage OA histopathology semi-quantitative scoring system (score 0–24)[36]. Subchondral bone grading (grade 0–3) was performed as previously described[70]. OA grading was scored by two blinded observers. For the muscle analysis, the knee joints of the sham (n = 4), negative control mimic (n = 5), and miR-455-5p/3p mimic groups (n = 5) were H&E-stained and the structural changes in muscle fibers, fibrosis, and infiltration of inflammatory cells were observed in the quadriceps femoris.

**Overexpression of miR-455-5p and -3p mimics in knee cartilage**. The target mimic-conjugated hydrogel was created as described previously[65]. In brief, 1.5 µl of miRNA mimic (50 µM), 1.5 µl of completion buffer, and 3 µl of invivofectamine 3.0 reagent (Thermo) were mixed, followed by incubation at 50 °C for 30 min. Then, 15 µl of PBS, 7.5 µl of AteloGene QG (KOKEN), and 7.5 µl of dilution buffer were added. The left knee was injected with the control miRNA mimic, and the right knee injected with miRNA-5p and -3p mimics in the knee joints of 6-month-old miR-455 knockout mice. Two days after injection, the knees were harvested and analyzed by immunohistochemistry. To confirm that the miRNA mimics reached the cartilage, we created an additional group that was injected with control miRNA or miR-455-5p/3p in 4-month-old miR-455 knockout mice. The efficacy of the miR-455 mimic was examined by qPCR from RNA extraction from cartilage 2 days after injection (Supplementary Fig. 14).

**Micro-CT scanning**. For micro-CT scanning, left femurs and skull were fixed in 70% ethanol for at least 1 week. Micro-CT scan was performed by using the microfocus CT system (InspeXio SMX-100CT; Shimadzu) under the following conditions: tube voltage, 60 kV; tube current, 40 µA; and field of view (XY), 10 mm. The resolution of one CT slice was 1024 × 1024 pixels and 9.0 µm/voxel. Bone morphometrics were analyzed by a 3D image analysis system (TRI/3D-BON; RATOC System Engineering, Tokyo, Japan). Parameters were calculated in 3D as follows: volumetric BMD (vBMD) was determined using a reference phantom (B-MAS200; KYOTO KAGAKU, Kyoto, Japan). Using a vBMD value for trabecular bone, >200 mg/cm$^3$ within the bone marrow was extracted. Bone volume (BV) was calculated using tetrahedrons corresponding to the enclosed volume of the triangulated surface. Total tissue volume (TV) analyzed was the entire marrow area volume, including trabecular bone. Trabecular bone volume fraction (BV/TV) was calculated from these values. Data are presented as the means + SD, and differences between groups were evaluated using the two-tailed Student's t test. A P value of <0.05 was considered statistically significant. Asterisk in figures indicates differences with statistical significance as follows: *P < 0.05.

**Immunohistochemistry**. For miR-455 KO analysis, joint sections were deparaffinized and activated with citric acid buffer (10 mM sodium citrate, 1 mM EDTA; pH 6.0) at 98–100 °C for 10 min in a microwave oven. After they were blocked with Blocking One solution (Nacalai tesque) for 30 min, they were incubated with rabbit anti-HIF-2α antibody (1:250; ab199, Abcam) overnight at 4 °C. After being this, they were incubated with rabbit Alexa Flour 488® donkey anti-rabbit antibody (1:500; A21206, Life Technologies) for 30 min, and mounted with VECTASHIELD mounting medium with DAPI (Vector Laboratories).

For the OA-model study, joint sections were deparaffinized, and the slides were washed and blocked with 10% goat serum for 1 h at room temperature. Anti-MMP13 (1:200; ab39012, Abcam) and anti-HIF-2α (1:300; ab20654, Abcam) were applied with 0.1% Tween 20 and incubated overnight at 4 °C and followed by secondary antibody using ImmPRESS reagents (Vector Laboratories). The signal was developed with 3,3wiDiaminobenzidine (DAB) with methyl green counterstaining.

**Skeletal preparation**. Neonatal (Postnatal day 0–2) miR-455$^{+/+}$ and miR-455$^{-/-}$ mice were fixed with 100% ethanol (EtOH) for 1 day after the majority of the skin and internal organs were removed. After fixation, the samples were incubated in Alcian blue solution (0.03% Alcian blue 8GX (SIGMA), 80% EtOH, 20% acetic acid) for overnight. The samples were rinsed with EtOH three times and incubated in Alizarin red solution (0.01% Alizarin Red S (SIGMA), 1% KOH) for overnight. The samples were treated with decoloring solution (1% KOH, 20% glycerol) for 6–10 days. The samples were then soaked in 20% glycerol, 20% EtOH solution.

**RNA isolation and quantitative RT-PCR**. Articular cartilage was collected from the femoral condyle and tibial plateau of each mouse as described previously[71]. Total RNA was isolated from chondrocytes and articular cartilage using TRIzol (Invitrogen), ISOGEN (Nippon gene), or TRI Reagent (Molecular Research Center) according to the manufacturer's protocol. Articular cartilage was collected from the femoral condyle and tibial plateau of each mouse, as described previously[71]. Synovium was collected from the anterior part of the knee joint. Total RNA was extracted using TRIzol (Invitrogen, Burlington, Ontario), followed by Zymo Direct-zol RNA MiniPrep kits (Zymo Research), according to the manufacturer's instructions. Complementary DNA (cDNA) was synthesized using ReverTra Ace (TOYOBO) or PrimeScript (Takara) with total RNA and random primers. Quantitative PCR was performed using THUNDERBIRD SYBR qPCR (TOYOBO) with gene-specific primers. Primer sequences are shown in Supplementary Table 3. The GAPDH gene was used as an internal control to normalize differences in each sample. miRNAs cDNA was synthesized using TaqMan MicroRNA Reverse Transcription Kit (Applied Biosystems) with total RNAs and each primer according to the manufacturer's protocol. Quantitative PCR was performed using TaqMan MicroRNA Assay probes for hsa-miR-455-5p (001280), hsa-miR-455-3p (002244), mmu-miR-455-3p (002455), hsa-miR-140-5p (001187), hsa-miR-140-3p (002234), hsa-miR-204 (000508), mmu-miR-204* (001199), hsa-miR-188-5p (002320), hsa-miR-188-3p (002106), hsa-miR-146b (001097), mmu-miR-146b* (002453), hsa-let-7a (000377), mmu-let-7a* (002478), hsa-miR-21 (000397), mmu-miR-21* (002493), hsa-miR-22 (000398), hsa-miR-22* (002301), hsa-miR-125b (000449), mmu-miR-125b* (002508), hsa-miR-199a-5p (000498), hsa-miR-199a-3p (002304), hsa-miR-133b (002247), mmu-miR-193b (002467), mmu-miR-193a-3p (002250), mmu-miR-19* (002577), hsa-miR-206 (000510), hsa-miR-378 (000567), hsa-miR-145 (002278), hsa-miR-29a* (002447), hsa-miR-133a (002246), hsa-miR-126 (002228), hsa-miR-1 (002222), hsa-miR-28 (000411), mmu-miR-28* (002545), RNU48 (001006), and snoRNA202 (001232). The RNU48 (for human samples) and snoRNA202 (for mouse samples) genes were used as an internal control to normalize differences in each sample.

**Plasmid construction and adenovirus infection**. pcDNA-miR-455 vector was constructed by inserting human pri-miR-455 sequence amplified by PCR using hsa-miR-455-F (5′-TTG AAT TCG CTT CCT TCT GCA GGT CCT GG-3′) and hsa-miR-455-R (5′-TTG ATA TCG CTC CTC CTC TTC CTC CCT G-3′) into pcDNA3.1 + (Invitrogen). We constructed CDS reporters by ligating PCR-amplified DNAs into the pLuc2-KAP-MCS vector[37]. Point-mutation reporters and 3′UTR reporters were carried out by inverse PCR of these reporters with phosphorylated primers and ligation of the amplicons. Primer sequences used for the inverse PCR are listed in Supplementary Table 4. Sox9-expressing adenovirus vector was constructed by inserting mouse Sox9 CDS into pAxCAwtit2 (TAKARA). The pAxCAiLacZit (TAKARA) was used as a LacZ-expressing adenovirus vector. The adenoviruses were produced by using adenovirus dual expression kit (TAKARA) according to the manufacturer's instructions and titered by 50% tissue culture infective dose (TCID$_{50}$) method. Adenovirus were infected into mouse primary chondrocytes at multiplicity of infections (MOIs) 5, 20, and 50.

**Identification of miR-455 targets by reporter library system**. Cell-based screening for the targets of miR-455 was performed using a reporter library, as previously described[37]. In brief, pcDNA-miR-455, or pcDNA3.1 vectors (Invitrogen), and a pRL-SV40 (Promega) *Renilla* luciferase construct were added to 384-well reporter library plates including 4.891 reporters with Opti-MEM containing FuGENE HD (Promega) and incubated for 20 min. Next, HEK293FT cells (Invitrogen) in 10% FBS/DMEM were added into each well. Cells were cultured in a 5% $CO_2$ incubator at 37 °C for 24 h. Then, wortmannin (final concentration of 10 nM, Sigma-Aldrich), a nonsense-mediated mRNA decay (NMD) inhibitor, was added into each well and cells were cultured in a 5% $CO_2$ incubator at 37 °C for a further 5 h. Luciferase activity was measured by ARVO (Perkin Elmer), using a Dual-Glo luciferase reporter assay system (Promega) and normalized to *Renilla* luciferase activity.

**Luciferase assay**. The HEK293FT cells in 96-well plates at 25% confluence were transfected using FuGENE HD (Promega) with luciferase reporter gene construct (25 ng), effector gene construct (50 ng), and 5 ng of pRL-SV40 (Promega) for normalization were co-transfected per well. Cells were cultured in a 5% $CO_2$ incubator at 37 °C for 24 h. Then, wortmannin (final concentration of 10 nM, Sigma-Aldrich) was added into each well, and cells were cultured in a 5% $CO_2$ incubator at 37 °C for a further 5 h. Luciferase activity was measured by ARVO (Perkin Elmer), using the Dual-Glo Luciferase Reporter Assay System (Promega).

**Epas1 knockdown in knee cartilage**. For the in vivo siRNA transfection experiment, control siRNA (siCtrl; 1022076) and siRNA against *Epas1* (siEpas1; SI04892867) were purchased from Qiagen (FlexiTube). 1.5 μl of siRNA (50 μM), 1.5 μl of completion buffer, and 3 μl of invivofectamine 3.0 reagent (Thermo) were mixed, followed by incubation at 50 °C for 30 min. Next, 15 μl of PBS, 7.5 μl of AteloGene QG (KOKEN), and 7.5 μl of dilution buffer were added. After 1 week of injection (every 3 days) into 6-month-old miR-455 KO mouse joints, mouse knee cartilages were harvested for qPCR analysis.

**Statistical analysis**. Statistically significant differences between two groups were evaluated using the two-tailed Student's *t* tests. Significant differences between more than two groups were analyzed by Dunnett's tests (one- or two-tailed) or Tukey–Kramer tests. Dunnett's tests and Tukey–Kramer tests were performed using RStudio.Version (3.5.1) (RStudio Team (2020). RStudio: Integrated Development for R. RStudio, PBC, Boston, MA URL http://www.rstudio.com/.). Differences were considered significant at $P < 0.05$ (*$P < 0.05$, **$P < 0.01$, ***$P < 0.001$).

**Reporting summary**. Further information on research design is available in the Nature Research Reporting Summary linked to this article.

## Data availability

All data supporting the findings described in this manuscript are available in the article and in the Supplementary Information, and from the corresponding author upon reasonable request. Source data are provided with this paper.

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

## Acknowledgements

We are grateful to all members of the Asahara Lab for their technical help and discussion. We thank S. Yamashita for his technical support in Sox9 downstream miRNA analysis and T. Kato for her technical support in generating the genome editing mice. We also appreciate Life Science Editors (https://www.lifescienceeditors.com) for English proof-reading support. This research was supported by the Japan Agency for Medical Research and Development (Core Research for Evolutional Science and Technology grants JP15gm0410001 and JP20gm0810008), the Naito Foundation, the Japan Society for the Promotion of Science (Kakenhi grants 18K19603, 19KK0227, 20H00547, and 17H04155), and National Institutes of Health, National Institute of Arthritis and Musculoskeletal and Skin Diseases (grants AR050631) (H.A.).

## Author contributions

Y.I., T.M., F.A., S.M., R.K., T.M., Y.T., M.I., H.T., L.L., and M.S. performed the experiments. Y.I., T.M., F.A., S.M., R.K., T.M., Y.T., and R.N. analyzed the data. S.M. created mouse figure. Y.I., T.M., F.A., K.N., M.L., and H.A. planned the experiments. Y.I., T.M., F.A., R.K., T.M., K.N., M.L., and H.A. wrote the manuscript.

## Competing interests

The authors declare no competing interests.
