## [Peer Review File · Nature Communications]

Reviewers' Comments:

Reviewer #1:

Remarks to the Author:

In this manuscript, the authors show that both isoforms of miR-455 are increased in cells over expressing the key chondrocyte transcription factor, Sox9. They demonstrate that global deletion of miR-455 promotes an age-related osteoarthritis (OA) phenotype, and 5 novel targets of miR-455 were identified, including Hif2a. The authors show that Hif2a is over expressed in cartilage from miR-455-null mice and that administration of miR-455 isoforms to the knee joint can attenuate OA and Hif2a expression in mice subjected to destabilization of the medial meniscus (DMM OA model).

While another group recently published a miR-455 knockout mouse model and demonstrated increased age-related OA, they focused on the novel target miR-455 target Pak2 and its interaction with the TGFB signaling pathway. Thus, the novel insights offered by the present paper include the intra-articular delivery of miR-455 isoforms as an OA therapeutic, the idea that both isoforms are regulated in parallel by Sox9, and that Hif2a is a novel target with potential impact on OA phenotype.

In general, the paper is well written and the data are pretty. However, missing in a lot of places is information that can be used to evaluate scientific rigor, including statistical analysis of the data, number of biological replicates and how many times each experiment was performed. This needs to be included for all figures, including the data from non-biased approaches.

At this time, specific comments on the data include:

1. The authors demonstrate that adenoviral-mediated delivery of miR-455 increases expression of Col2a1 in mouse chondrocytes in vitro. They also demonstrate the level of over expression achieved by the adenovirus. However, in order to put these data into context, it will be important for the authors to show the time course and level of miR-455 regulation seen in chondrocytes differentiating in vivo, in a pellet or micromass type culture. Correlation of miR-455 levels with Sox9 and differentiation markers would be very helpful.
2. When discussing the ChiP data, the authors should state in the text where, in relation to the miR-455 locus, are the elements of interest.
3. The authors demonstrate the impact of intra-articular miR-455 isoform delivery on OA phenotype. Missing are data on whether this treatment protocol has an impact articular cartilage phenotype in the absence of injury.
4. The authors have missed the opportunity to show whether or not the miR-455 delivery is impacting only the intra-articular space or whether it has impacts on surrounding bone or skeletal muscle, as miR-455 is also active in these compartments.
5. The authors need to be more forthcoming in the discussion, and possibly the introduction, about the fact that a miR-455-null mouse has already been published and that it displays an age-related OA phenotype similar to that of their own mouse model. The goal of a discussion is to discuss the data presented and how it may be considered/integrated/related to what is already known on the topic. Giving fair consideration to previously published work does not diminish a strong data set.

Reviewer #2:

Remarks to the Author:

The manuscript by Ito Y. et al reported that both miR-455-5p and -3p, up-regulated by Sox9, are important players involved in OA pathogenesis. Overexpression of both miR-455-5p and -3p protected cartilage degeneration in a mouse OA model. It is interesting to report that both -5P and -3p derived from one miRNA play similar roles. However, previous literature reported a similar role of miR-455-3p in chondrocyte differentiation and OA. This ms found that -5p and -3p target same targets, which does not include PAK2 that is a key target of -3p in ref38. This ms does not thoughtfully clarify the underlying mechanisms by which miR455 functions. In addition, how does miR455 act by targeting multiple targets to regulate OA pathogenesis? which one or few are the key targets? If Hif2a is the key one, then rescue of miR455 KO phenotype by Hif2a deletion should be tested. What are off-targeting effects when using miR455 mimics in vivo? What are the other organs' phenotypes in miR455 KO mice because these mice are global KO mice? Overall, this ms reported partial phenotype of miR455 KO mice, but the mechanisms by which miR455 regulate chondrocytes and OA are far from well explored. The other important issues for this ms include critical defects in statistical analysis (ttest is not the right method for analyzing data with more than 2 groups. This ms only used that method, which is a big defect that dampens data reliability. Moreover, there are no n numbers throughout the ms except for the OA model.) and less details of methods, which is hard for others to reproduce their work.

Essential methodology is missing:

The detailed methodology of increasing expression of Hif-2a in 6-mo miR-455 knockout cartilage in Fig4 e,f is unclear.

Details of OARSI, and injection of keen are not clear.

Overexpression extent of miR455 in Fig4, 5 should be shown.

Fig. 1d, As the authors' data show that miR-455 produces almost the same amount of both the 5p and 3p strands in chondrocytes, why there is no significance in miR455-3p expression in OA cartilage?

Fig. 4C,E,F, Fig5 B,D, the corresponding images under light microscope should be provided.

In addition MMP13, cartilage degeneration-related genes should be characterized.

The location of miR455 in eFig 2a should be labeled.

Dotted figure should be shown in eFig 3.

Reviewer #3:

Remarks to the Author:

Major Comments:

1. Figure1b. Why did the authors only verify miR140/455, and the descending miRs also need to be verified.
2. Figure1e, the indicators such as COL2A1 and MMP13, should be added in the IL-1beta stimulus model.
3. Figure2d, it is suggested that PCR/WB/immunohistochemistry of COL2A1,MMP13,and SOX9 may be performed to compare the difference in 2/6 month mice, and macroscopic scoring and Safranin-O staining were not enough.
4. The author made the miR-455/HIF-2a axis. But the reviewer suggested in-depth discussion of the downstream mechanism of HIF-2a on cartilage, which would make the article better.

Minor Comments:

- 1.The number of experiments should be added in the Figure Legend.

Reviewers' comments:

Reviewer #1 (Remarks to the Author):

In this manuscript, the authors show that both isoforms of miR-455 are increased in cells over expressing the key chondrocyte transcription factor, Sox9. They demonstrate that global deletion of miR-455 promotes an age-related osteoarthritis (OA) phenotype, and 5 novel targets of miR-455 were identified, including Hif2a. The authors show that Hif2a is over expressed in cartilage from miR-455-null mice and that administration of miR-455 isoforms to the knee joint can attenuate OA and Hif2a expression in mice subjected to destabilization of the medial meniscus (DMM OA model).

While another group recently published a miR-455 knockout mouse model and demonstrated increased age-related OA, they focused on the novel target miR-455 target Pak2 and its interaction with the TGF β signaling pathway. Thus, the novel insights offered by the present paper include the intra-articular delivery of miR-455 isoforms as an OA therapeutic, the idea that both isoforms are regulated in parallel by Sox9, and that Hif2a is a novel target with potential impact on OA phenotype.

In general, the paper is well written and the data are pretty. However, missing in a lot of places is information that can be used to evaluate scientific rigor, including statistical analysis of the data, number of biological replicates and how many times each experiment was performed. This needs to be included for all figures, including the data from non-biased approaches.

We appreciate this important point. We have carefully reviewed the MS and now state the significant differences in the Figures, and have ensured that the number of samples and experiments are described in all Figure legends. Changes in the revised text are marked in red.

At this time, specific comments on the data include:

1. The authors demonstrate that adenoviral-mediated delivery of miR-455 increases expression of Col2a1 in mouse chondrocytes in vitro. They also demonstrate the level of over expression achieved by the adenovirus. However, in order to put these data into context, it will be important for the authors to show the time course and level of miR-455

regulation seen in chondrocytes differentiating in vivo, in a pellet or micromass type culture. Correlation of miR-455 levels with Sox9 and differentiation markers would be very helpful.

Thank you for the suggestion. We have now analyzed miR-455s expression during chondrogenesis of mouse MSCs in pellet cultures. We have added the following text and data in the revised MS.

RESULTS section, “miR-455s expression are regulated by Sox9 in chondrocytes”:

In addition, we measured miR-455s expression during chondrogenesis of mouse MSCs in pellet cultures. The expression of both miR-455-5p and -3p was increased in 1 week differentiated cells compared to 0 week (Extended Data Fig. 4). The peaks of early chondrogenesis genes such as *Col2a1* and *Acan* peaked at 2 weeks of differentiation, whereas miR-455 had an early peak at the first week (Extended Data Fig. 4), indicating that miR-455 is expressed at the early stage of chondrogenesis.

Extended Data Fig. 4. Relative expression levels of miR-455s and cartilage marker genes during mouse MSC chondrogenesis in pellet cultures. Error bars show SEM, n=3.

2. When discussing the ChIP data, the authors should state in the text where, in relation to the miR-455 locus, are the elements of interest.

The position of the miR-455 locus has been added in Extended Data Fig. 2a. Also, we carefully rewrote the ChIP data section of the Results as follows:

RESULTS, “miR-455s expressions are regulated by Sox9 in chondrocytes”:

To investigate whether expression of miR-455 is directly regulated by Sox9, we performed chromatin immunoprecipitation (ChIP) analysis for Sox9 in mouse chondrocytes. miR-455 is located in intron 10 of *Col27a1*. Intronic miRNAs are believed to be processed from the introns of their hosting transcription units and hence share common regulatory mechanisms and expression patterns with the host gene²⁸⁻³⁰. Since cartilage is a tissue conserved in mammals, we considered that the Sox9-binding sequence exists in the genomic region conserved among mammals. To search the transcriptional regulating region of *Col27a1*, we first investigated conservation of the genomic sequence near the transcriptional start site of *Col27a1* (-20K to +20K). The genomic sequence was compared with that of human, chimpanzee, rhesus, cow and dog by VISTA-point³¹. We identified 16 conserved regions and created 19 primer sets for the ChIP (Extended Data Fig. 2a and Extended Data Table 1). ChIP using anti-Sox9 antibody in mouse primary chondrocytes revealed that Sox9 binds a conserved region of *Col27a1* intron 3 (primer set no. 16) (Extended Data Fig. 2b). Previous reports indicated that Sox9 homodimers bind to enhancer regions that contain inverted Sox9 binding sites separated by 3–4 bp and represent a palindromic motif for chondrocyte-specific gene regulation³²⁻³⁴. This region contained a palindromic sequence similar to the Sox9-binding consensus sequence separated by 4 bp (Extended Data Fig. 2c). These data suggest that miR-455 and the host gene *Col27a1* are directly regulated by Sox9 in chondrocytes.

Extended Data Fig. 2. ChIP analysis using anti-Sox9 antibody on the *Col27a1* gene locus in mouse chondrocytes.

(a) Conservation analysis of the mouse *Col27a1* gene locus by VISTA-point, and positions of the primer sets for the ChIP analysis (below).

(b) Quantitative ChIP analysis using anti-Sox9 antibody in mouse chondrocytes. Error bars show SEM. **This assay was performed independently three times (n=3).**

(c) Sox9-binding consensus-like sequence of primer set no. 16 region in *Col27a1* intron 3. Mo, mouse; Hu, human; Ch, chimpanzee; Rh Rhesus; Co, cow; Do, dog.

3. The authors demonstrate the impact of intra-articular miR-455 isoform delivery on OA phenotype. Missing are data on whether this treatment protocol has an impact articular cartilage phenotype in the absence of injury.

We agree with the reviewer's comment, and have additionally performed miR-455 delivery in the absence of injury. We found that injection of the miRNA mimic did not affect the

cartilage and surrounding tissue phenotype. We have added the following text in the Results and data in Extended Data Fig. 11.

RESULTS, “Introduction of both miR-455-5p and -3p inhibits cartilage degeneration”:
We first investigated whether introduction of miRNA mimics had an impact on articular cartilage and the phenotype of the surrounding tissue in the absence of DMM surgery. The cartilage degeneration score and subchondral bone score of the negative control mimic- or miR-455s-introduced group did not differ from those of the sham group (Extended Data Fig. 11a-c). Regarding the surrounding muscles (quadriceps femoris), no difference was observed between the sham group and the miRNA-transfected group (Extended Data Fig. 12). These data indicate that injection of the miRNA mimic does not affect the phenotype of the knee joint.

Extended Data Fig. 11. Phenotype analysis of sham or miRNA control mimic (miNega) or miR-455-5p/3p (miR-455) injected-knee joint.

(a) The OARSI scores of sham or miRNA control mimic (miNega) or miR-455-5p/3p (miR-455) injected knee joints. Error bars show SEM, n=6~8. ns: not significant.

(b) The subchondral bone scores of sham or miRNA control mimic (miNega) or miR-455-5p/3p (miR-455) injected knee joints. Error bars show SEM, n=6~8. ns: not significant.

(c) Representative image of Safranin O staining of miRNA mimic-injected knee joints. Scale bar shows 200 μ m.

4. The authors have missed the opportunity to show whether or not the miR-455 delivery is impacting only the intra-articular space or whether it has impacts on surrounding bone or skeletal muscle, as miR-455 is also active in these compartments.

As mentioned above, we also investigated the subchondral bone and quadriceps femoris. The above text and data, and the following Figure have been added to the revised MS.

Extended Data Fig. 12. Representative image of H&E staining for quadriceps of sham or miRNA control mimic (miNega) or miR-455-5p/3p (miR-455) injected knee joint. Scale bar shows 200 μ m.

5. The authors need to be more forthcoming in the discussion, and possibly the introduction, about the fact that a miR-455-null mouse has already been published and that it displays an age-related OA phenotype similar to that of their own mouse model. The goal of a discussion is to discuss the data presented and how it may be considered/integrated/related

to what is already known on the topic. Giving fair consideration to previously published work does not diminish a strong data set.

Thank you very much for this important point. We rewrote parts of the Introduction and Discussion as follows:

INTRODUCTION, paragraph 4:

In the present study, we demonstrate that the expressions of both miR-455-5p and -3p from the same precursor miR-455 (pre-miR-455) are upregulated by overexpression of Sox9. Recently, it has been reported that aged miR-455-3p knockout mice show significant degeneration of knee cartilage, indicating that miR-455-3p is a critical cartilage homeostasis regulator²⁷. However, the mechanisms of miR-455 functions in cartilage homeostasis remain largely unknown. In the present study, we generated miR-455 knockout mice and confirmed an age-related OA-like phenotype similar to the previous report. We also investigated the targets of miR-455s using a reporter library system, which is a cell-based screening system for targets of post-transcriptional regulators. This system identified Hif-2 α , a major catabolic factor for cartilage, as a target of both miR-455-5p and -3p. The cartilage degeneration in miR-455 knockout mice is partially rescued by knockdown of Hif-2 α expression. Moreover, to examine the therapeutic effect of miR-455s, we performed overexpression of miR-455s in OA-model mice. The results strongly implicate miR-455-5p and -3p in supporting articular cartilage homeostasis by targeting Hif-2 α .

DISCUSSION, paragraph 4:

The miR-455 knockout mice showed an OA-like phenotype, which is consistent with a recent report of 5-mo miR-455-3p deletion mice²⁷. The mice in that report were generated by partial deletion of the miR-455-3p sequence, which is expected to inhibit the formation of the pre-miR-455 structure, and thus inhibit expression of both miR-455-5p and -3p. This report also indicated that miR-455-3p directly targets *PAK2*, which they showed was an inhibitor of TGF- β /Smad signaling that plays a key role in maintaining cartilage homeostasis²⁷. They also indicated that the expression of *PAK2* is upregulated in OA cartilage²⁷. *PAK2* knockdown as well as overexpression of miR-455-3p in human OA chondrocytes correlated with increased expression of cartilage-specific genes *SOX9*, *COL2A1* and *ACAN*, and decreased expression of cartilage degeneration-related genes *RUNX2*, *COL10A1* and *MMP13*²⁷. Thus, *PAK2* and HIF-2 α are thought to be critical genes for cartilage degeneration. Targeting these

two genes by miR-455 might contribute more effectively to maintain cartilage homeostasis. We propose that the suppression of different genes that promote cartilage degeneration by two miRNAs generated from one pre-miRNA more effectively controls cartilage homeostasis.

Reviewer #2 (Remarks to the Author):

The manuscript by Ito Y. et al reported that both miR-455-5p and -3p, up-regulated by Sox9, are important players involved in OA pathogenesis. Overexpression of both miR-455-5p and -3p protected cartilage degeneration in a mouse OA model. It is interesting to report that both -5P and -3p derived from one miRNA play similar roles. However, previous literature reported a similar role of miR-455-3p in chondrocyte differentiation and OA. This ms found that -5p and -3p target same targets, which does not include PAK2 that is a key target of -3p in ref38. This ms does not thoughtfully clarify the underlying mechanisms by which miR455 functions. In addition, how does miR455 act by targeting multiple targets to regulate OA pathogenesis? which one or few are the key targets? If Hif2a is the key one, then rescue of miR455 KO phenotype by Hif2a deletion should be tested.

We would like to thank the reviewer for his/her careful review of our manuscript and the questions raised. It is of course important to consider our work in the context of the related work recently published, and compare/discuss it accordingly. We did not detect PAK2 as a target of miR-455 because our reporter library, which includes about 5,000 genes, does not include PAK2. In this study, we focused our experiments on what we did detect as a target of miR-455-5p and -3p, , i.e., Hif-2 α , so we only discuss PAK2 in the Discussion.

We have now added further data to help clarify the underlying mechanism of miR-455 function. Specifically, we tested whether the phenotypes associated with cartilage degeneration in the miR-455 knockout mice are due to the increased expression of *Epas1* in the cartilage, by performing *Epas1* knockdown by siRNA in 6-month-old KO mouse knee cartilage. We note that a double knockout of miR-455 and *Epas1* could have been a more ideal experiment. However, given the current pandemic and the limitations that has brought to the laboratories, this would be very difficult for us to perform, and anyway would take at least XX months even if we did have full access. We also note that it may not even be technically possible because the *Epas1* is a critical gene for development and *Epas1* KO mice are embryonic lethal. Therefore, we chose the *in vivo* knockdown system in knee cartilage of miR-455 KO mice as an experiment that we could perform, and a good first step to help clarify the mechanism. Knockdown of *Epas1* in articular cartilage of miR-455 knockout mice rescues the abnormal increased expression of cartilage degeneration-related genes, *Mmp3* and *Mmp13*, which have been reported to be transcriptional targets of *Epas1*. The following text and Figure have been added to the revised MS.

RESULTS, “**Knockdown of *Epas1* in miR-455 knockout knee cartilage rescues the abnormal increased expression of cartilage degeneration-related genes**”:

To investigate whether the phenotypes associated with cartilage degeneration in miR-455 knockout mice are due to the increased expression of *Epas1* in the cartilage, we performed *Epas1* knockdown by siRNA in 6-month-old KO mouse knee cartilage (Fig. 6a). The expression of *Epas1* in knee cartilage of 6-month-old wild-type and KO mice was significantly reduced by knockdown of *Epas1* (Fig. 6b). Moreover, the expression of *Mmp3* and *Mmp13*, which have been reported to be downstream target genes of Hif-2 α ^{41, 42}, were significantly decreased by *Epas1* knockdown (Fig. 6b). Similarly, the expression of *Nos2*, which is also considered to be a downstream target gene of Hif-2 α ⁴², and a cartilage degeneration marker gene, *Adamts5*, were decreased by *Epas1* knockdown in miR-455 KO knee cartilage (Fig. 6b), although no significant differences were observed (P=0.0573, and P= 0.0569, respectively). These data indicate that the phenotype of cartilage degeneration in miR-455 knockout mice is due, at least in part, to the increased expression of *Epas1* in the cartilage.

Fig. 6. Knockdown of *Epas1* in miR-455 KO knee cartilage rescues phenotypes associated with cartilage degeneration.

(a) Schedule of treatment using transfection of siRNA for *Epas1* in knee cartilage of 6-month-old miR-455 knockout mice.

(b) Relative expression levels of *Epas1*, *Mmp3*, *Mmp13*, *Adamts5* and *Nos2* in siRNA-transfected knee cartilage of 6-month-old WT or miR-455 knockout mice. Error bars show SEM, n=5.

What are off-targeting effects when using miR455 mimics in vivo?

This is an important point, and we have now additionally performed miR-455 delivery in the absence of injury to test for potential off-target effects that may influence our results. We found that injection of the miRNA mimic had no effect on the cartilage and subchondral bone phenotype. We have added the following text in the Results and data in Extended Data Fig. 11.

RESULTS, “Introduction of both miR-455-5p and -3p inhibits cartilage degeneration”: We first investigated whether introduction of miRNA mimics had an impact on articular cartilage and the phenotype of the surrounding tissue in the absence of DMM surgery. The cartilage degeneration score and subchondral bone score of the negative control mimic- or miR-455s-introduced group did not differ from those of the sham group (Extended Data Fig. 11a-c). Regarding the surrounding muscles (quadriceps femoris), no difference was observed between the sham group and the miRNA-transfected group (Extended Data Fig. 12). These data indicate that injection of the miRNA mimic does not affect the phenotype of the knee joint.

Extended Data Fig. 11. Phenotype analysis of sham or miRNA control mimic (miNega) or miR-455-5p/3p (miR-455) injected-knee joint.

(a) The OARSI scores of sham or miRNA control mimic (miNega) or miR-455-5p/3p (miR-455) injected knee joints. Error bars show SEM, n=6~8. ns: not significant.

(b) The subchondral bone scores of sham or miRNA control mimic (miNega) or miR-455-5p/3p (miR-455) injected knee joints. Error bars show SEM, n=6~8. ns: not significant.

(c) Representative image of Safranin O staining of miRNA mimic-injected knee joints. Scale bar shows 200 μ m.

Extended Data Fig. 12. Representative image of H&E staining for quadriceps of sham or miRNA control mimic (miNega) or miR-455-5p/3p (miR-455) injected knee joint. Scale bar shows 200 μ m.

What are the other organs' phenotypes in miR455 KO mice because these mice are global KO mice?

This is a good point, and we have now analyzed other organs of the miR-455 KO mice and have added the following text to the Results, and data in Extended Data Fig. 9-10, and Extended Data Table 2.

RESULTS, "OA-like pathology is observed in miR-455 knockout mouse knee joints":

The miR-455^{-/-} mice were born normally, and the body weight of post-neonatal day 0 (P0) and 4-week-old mice was similar to wild type mice (Extended Data Fig. 7). Because chondrocytes play a critical role in skeletal development, we investigated skeletal formation in miR-455^{-/-} mice by performing skeletal prep with neonatal miR-455^{-/-} and wild-type mice. Skeletal development was normal in miR-455 knockout mice (Extended Data Fig. 8). We also performed micro-CT scanning in 8-week-old male miR-455^{-/-} mice (Extended Data Fig. 9). There were no significant changes in bone formation of the femur and skull in miR-455^{-/-} mice. Bone volume and mineral density in femur were not significantly changed in miR-455^{-/-} mice. In addition, gross tissue size analysis and histological analysis for major tissues (thyroid, pancreas, spleen, heart, lung, kidney, liver and testis) were performed in 8-week-old miR-455 knockout mice, but no difference was observed between wild-type and knockout mice (Extended Data Fig. 10 and Extended Data Table 2).

Extended Data Fig. 9. Micro-CT analyses of wild-type or miR-455 knockout mice (8-week-old).

(a) Micro-CT images of femur in wild-type and miR-455 knockout mice. scale bars show 1 mm.

(b) Trabecular bone volume fraction (BV/TV) and volumetric BMD (vBMD) of cancellous bone and tissue mineral density (TMD) of trabecular bone in wild-type and miR-455 knockout femur. Error bars represent SD, n=3.

(c) Micro-CT images of skull in wild-type and miR-455 knockout mice. scale bars show 10 mm.

(d) Skull length of wild-type and miR-455 knockout mice. Error bars represent SD, n=3.

Extended Data Fig. 10. Representative images of H&E staining for major tissues of wild-type and miR-455 knockout mice (8-week-old). Scale bar shows 600 μ m.

		WT1	WT2	WT3	KO1	KO2	KO3	Average		SD		p value
								WT	KO	WT	KO	
Height	mm	88.27	87.5	88.52	87.96	87.86	88.76	88.1	88.19	0.43	0.4	0.83
Weight	g	27.3	26.7	27.6	29	27.3	30.6	27.2	28.97	0.37	1.35	0.15
Subcutaneous fat	mm	0.4	0.44	0.44	0.49	0.46	0.46	0.43	0.47	0.02	0.01	0.06
Lower limb length	mm	37.54	37.64	39.15	39.54	37.44	37.58	38.11	38.19	0.74	0.96	0.93
Thickness of the quadriceps muscle	mm	3.69	3.82	3.67	3.62	3.72	3.63	3.73	3.66	0.07	0.04	0.29
Tooth length	mm	4.19	4.3	4.16	4.16	3.97	4.01	4.22	4.05	0.06	0.08	0.08
Abdominal girth	mm	68.33	67.02	64.03	69.63	68.97	69.47	66.46	69.36	1.8	0.28	0.09
Thyroid	g	0.17	0.31	0.22	0.26	0.28	0.38	0.23	0.31	0.06	0.05	0.26
Lung	g	0.29	0.34	0.34	0.34	0.36	0.45	0.32	0.38	0.02	0.05	0.19
Heart	g	0.13	0.15	0.13	0.14	0.14	0.15	0.14	0.14	0.01	0	0.42
Liver	g	2.2	2.05	2.28	2.38	1.76	2.4	2.18	2.18	0.1	0.3	0.99
Pancreas	g	0.14	0.23	0.27	0.16	0.2	0.19	0.21	0.18	0.05	0.02	0.5
Spleen	g	0.17	0.16	0.16	0.16	0.18	0.16	0.17	0.17	0	0.01	0.68
Kidney (Rt)	g	0.26	0.29	0.31	0.36	0.27	0.32	0.29	0.32	0.02	0.04	0.37
Tests	g	0.2	0.24	0.16	0.2	0.13	0.11	0.2	0.15	0.03	0.04	0.21

Overall, this ms reported partial phenotype of miR455 KO mice, but the mechanisms by which miR455 regulate chondrocytes and OA are far from well explored.

As indicated above, we have added some important data to address this point in the revised MS. The miR-455 KO mice showed OA-like cartilage degeneration at 6-mo. We identified Epas1 (Hif-2 α) as a target of miR-455-5p and -3p. The expression of Epas1 and its transcriptional targets, the cartilage degenerative genes Mmp3 and Mmp13 were significantly increased in 6-mo miR-455 KO cartilage. Overexpression of miR-455-5p and -3p inhibited the cartilage destruction in surgical OA model mice. Furthermore, knockdown of Epas1 in 6-mo miR-455 KO cartilage rescues the elevated expression of Mmp3 and Mmp13, downstream genes of Epas1. These new data now provide further support that miR-455-5p and -3p regulate cartilage homeostasis at least in part by targeting Epas1.

The other important issues for this ms include critical defects in statistical analysis (ttest is not the right method for analyzing data with more than 2 groups. This ms only used that method, which is a big defect that dampens data reliability. Moreover, there are no n numbers throughout the ms except for the OA model.) and less details of methods, which is hard for others to reproduce their work.

We appreciate the reviewer's comment. Statistical analysis of all our data only analyzes the differences between two groups, therefore all statistical analyses were performed with a 2-tailed Student's t-test. However, we appreciate that the lack of description, and have carefully reviewed the MS and now state the significant differences in the Figures, and the number of the samples and experiments in the Figure legends. All changes in the text are marked in red.

Essential methodology is missing:

The detailed methodology of increasing expression of Hif-2a in 6-mo miR-455 knockout cartilage in Fig4 e,f is unclear.

We apologise and have rewritten the method as follows:

Overexpression of miR-455-5p and -3p mimics in knee cartilage

The target mimic conjugated hydrogel was created as described previously⁶⁴. In brief, 1.5 μ l miRNA mimic (50 μ M), 1.5 μ l completion buffer, and 3 μ l invivoFectamine 3.0 reagent (Thermo) were mixed, followed by incubation at 50 °C for 30 minutes. Then, 15 μ l PBS, 7.5 μ l AteloGene QG (KOKEN), and 7.5 μ l dilution buffer were added. The

left knee was injected with the control miRNA mimic, and the right knee injected with miRNA-5p and -3p mimics in the knee joints of 6-month-old miR-455 knockout mice. Two days after injection, the knees were harvested and analyzed by immunohistochemistry. To confirm that the miRNA mimics reached the cartilage, we created an additional group that were injected with control miRNA or miR-455-5p/3p in 4-month-old miR-455 knockout mice. The efficacy of the miR-455 mimic was examined by qPCR from RNA extraction from cartilage 2 days after injection (Extended Data Fig. 14).

Details of OARSI, and injection of knee are not clear.

We apologise for the lack of clarity and have rewritten the method as follows:

OA-model mice

C57BL/6J mice were used for this study. All animal procedures were performed according to protocols approved by the Institutional Animal Care and Use Committee (IACUC) at The Scripps Research Institute. All mice were freely allowed to access to food, water, and activity. In the treatment study, OA was surgically induced by DMM in the right knee joints of 3-mo mice⁶⁸. The left knees were subjected to sham surgery. The target mimic conjugated hydrogel was created as described previously⁶⁴. In brief, 1.5 µl miRNA mimic (50 µM), 1.5 µl completion buffer, and 3 µl invivoFectamine 3.0 reagent (Thermo) were mixed, followed by incubation at 50 °C for 30 minutes. Then, 15 µl PBS, 7.5 µl AteloGene QG (KOKEN), and 7.5 µl dilution buffer were added. The total volume for injection was 30 µl. The mice (n=48) were divided into four groups. Group 1 (control group) was treated with control miRNA mimic (Pre-miR, Ambion). Group 2 was treated with mouse miR-455-3p (Ambion). Group 3 was treated with mouse miR-455-5p (Ambion). Group 4 was treated with miR-455-3p and miR-455-5p. Two days after DMM, the treatment of mimic with an injection to the right knee started 2 times a week for 10 weeks and the mice were killed at 10 weeks after surgery. The entire knee joints were fixed in 10% zinc-buffered formalin for 2 days and decalcified in TBD-2 for 24 h. To investigate whether the mimic reached the cartilage, we created two additional groups that were treated with only mimic injection 2 times a week for 10 weeks. One was treated with control miRNA mimic, and the other with miR-455-5p and -3p (n=3, each). The efficacy of the miR-455 mimic was examined by qPCR from RNA extraction from cartilage 3 days after the final injection (Extended Data Fig. 12). To test

whether the injection of the miRNA mimic affects the phenotype, we created three additional groups that were treated only with mimic injection 2 times a week for 10 weeks. Group 1 was treated with control miRNA mimic (n=6), Group 2 with miR-455-5p and -3p (n=8), and group 3 was subjected to sham surgery (n=6). The entire knee joints were fixed in 10% formalin for a day and decalcified in formic acid/sodium citrate solution for 24 h. The samples were embedded and stained with Safranin-O.

Histological analysis

For H&E staining, tissues from 8-week-old male mice were fixed in 4% paraformaldehyde/0.1 M phosphate buffer, and embedded in paraffin. Sections of 4 μ m in thickness were stained with hematoxylin (131-09665, FUJIFILM Wako Pure Chemical Corp.) and eosin (051-06515, FUJIFILM Wako Pure Chemical Corp.).

Mouse knee joints were harvested, fixed, decalcified, embedded and stained with Safranin-O. The histological OA scores for medial femoral condyle, the medial tibial plateau, and summed scores of femur and tibia were evaluated using the Osteoarthritis Research Society International (OARSI) cartilage OA histopathology semi-quantitative scoring system (score 0–24)³⁶. Subchondral bone grading (grade 0-3) was performed as previously described⁶⁹. OA grading was scored by two blinded observers.

Overexpression extent of miR455 in Fig4, 5 should be shown.

The following data were added to Extended Data Fig 13-14.

Extended Data Fig. 13. Relative expression of miR-455s in negative control mimic (Ctrl) or miR-455s mimic (miR-455s) transfected articular cartilage of C57BL6 mice. Error bars indicate SEM, n=3.

Extended Data Fig. 14. Relative expression of miR-455s (normalized to snoRNA202) in negative control mimic (Ctrl) or miR-455s mimic (miR-455s) transfected articular cartilage of 4-month-old miR-455 knockout mice. Error bars indicate SEM, n=3. ND: not detected.

Fig. 1d, As the authors' data show that miR-455 produces almost the same amount of both the 5p and 3p strands in chondrocytes, why there is no significance in miR455-3p expression in OA cartilage?

It has been reported that the ratio of the two mature strands of many miRNA duplexes can vary depending on cell type, developmental stage and in several diseases (reviewed in *Biochem. Soc. Trans.* **2014**, *42*, 1135–1140). Therefore, it is not unexpected that the ratios of 5p and 3p strands differ between samples in different denatured states. The regulation of miRNA strand selection is a very interesting topic, but we feel it is outside the scope of the present paper.

Fig. 4C,E,F, Fig5 B,D, the corresponding images under light microscope should be provided.

Although the images in Fig. 5b and d are bright field images, Fig. 4c and e were replaced as follows:

- (c) Representative image of Hif-2 α , DAPI staining and bright field of wild-type and miR-455 knockout articular cartilage (2, 6-month-old). Scale bars show 100 μ m.
- (d) The percentage of Hif-2 α positive cells. Error bars show SD, n=4.
- (e) Representative image of Hif-2 α and DAPI staining and bright field view of 6-mo wild-type and miR-455 knockout articular cartilage transfected with control mimic (Ctrl) or miR-455-5p and -3p mimics (miR-455-5p/3p). Scale bars show 100 μ m.
- (f) The percentage of Hif-2 α positive cells. Error bars show SD, n=4.

In addition MMP13, cartilage degeneration-related genes should be characterized.

Thank you for this valuable suggestion. We added the following data and identified that expression of cartilage degeneration-related genes were increased in 6-mo miR455 KO mice.

Fig. 2. miR-455 knockout mice show OA-like phenotype.

(d) Relative mRNA levels of cartilage markers and cartilage degeneration related genes in knee cartilage of wild-type and miR-455 knockout mice (2, 6-month-old). Error bars show SEM, n=5.

The location of miR455 in eFig 2a should be labeled.

The Figure was replaced as follows:

Extended Data Fig. 2. ChIP analysis using anti-Sox9 antibody on the *Col27a1* gene locus in mouse chondrocytes.

(a) Conservation analysis of the mouse *Col27a1* gene locus by VISTA-point, and positions of the primer sets for the ChIP analysis (below).

(b) Quantitative ChIP analysis using anti-Sox9 antibody in mouse chondrocytes. Error bars show SEM. **This assay was performed independently three times (n=3).**

(c) Sox9-binding consensus-like sequence of primer set no. 16 region in *Col27a1* intron 3. Mo, mouse; Hu, human; Ch, chimpanzee; Rh Rhesus; Co, cow; Do, dog.

Dotted figure should be shown in eFig 3.

The Figure was replaced as follows:

Extended Data Fig.3. Relative mRNA levels of miR-455-5p (455-5p) and -3p (455-3p) in human MSCs (n=7) or chondrocytes (n=10). Error bars show SEM.

Reviewer #3 (Remarks to the Author):

Major Comments:

1. Figure1b, Why did the authors only verify miR140/455, and the descending miRs also need to be verified.

We appreciate the point. Fig. 1b has now been replaced as follows:

Fig. 1. Expressions of miR-455-5p and -3p in chondrocytes.

(b) Relative expression of miRNAs in Sox9- or LacZ-expressing adenovirus-infected chondrocytes. Error bars show SEM, n=3. ns: not significant.

2. Figure1e, the indicators such as COL2A1 and MMP13, should be added in the IL-1beta stimulus model.

The expression data of *Mmp13* and *Col2a1* has now been added.

Fig. 1. Expressions of miR-455-5p and -3p in chondrocytes.

(e) Relative expression of pri-miR-455, Col27a1, mature-miR-455s, Sox9, Col2a1 and Mmp13 in mouse primary chondrocytes with or without IL-1 β stimuli. Error bars show SEM, n=3.

3. Figure2d, it is suggested that PCR/WB/immunohistochemistry of COL2A1, MMP13, and SOX9 may be performed to compare the difference in 2/6 month mice, and macroscopic scoring and Safranin-O staining were not enough.

Thank you for your comment. We have added the following data and identified that cartilage degeneration-related genes were increased in 6-mo miR455 KO mice.

Fig. 2. miR-455 knockout mice show OA-like phenotype.

(d) Relative mRNA levels of cartilage markers and cartilage degeneration related genes in knee cartilage of wild-type and miR-455 knockout mice (2, 6-month-old). Error bars show SEM, n=5.

4. The author made the miR-455/HIF-2 α axis. But the reviewer suggested in-depth discussion of the downstream mechanism of HIF-2 α on cartilage, which would make the article better.

We appreciate this point and have rewritten the discussion as follows:

DISCUSSION, paragraph 3-4:

HIF-2 α is a homolog of HIF-1 α and a member of the basic helix-loop-helix/PAS transcription factor family⁵⁵. It has been implicated in osteoarthritis pathogenesis. HIF-2 α has an abnormal expression pattern in OA cartilage and regulates hypertrophic differentiation of OA chondrocytes⁴¹. *Epas1*^{+/-} mice show significant resistance to cartilage destruction and overexpression of Hif-2 α enhances cartilage degradation^{41,42}. Hif-2 α directly promotes the expression of catabolic genes such as *Mmp3*, *Mmp13*, *Col10a1* and *Nos2*^{41,42}. These data indicate that HIF-2 α causes cartilage destruction by transactivating catabolic genes. In this study, we demonstrated that miR-455s directly

regulates HIF-2 α expression. In addition, we also find that miR-455 knockout mice at 6-mo showed destruction of knee cartilage and enhanced expression of Hif-2 α and its target genes *Mmp3* and *Mmp13* in knee cartilage. Moreover, introduction of miR-455-5p and -3p mimics prevented DMM-induced cartilage destruction and reduced Hif-2 α expression. Furthermore, knockdown of *Epas1* in articular cartilage of miR-455 knockout mice rescues the abnormal increased expression of the cartilage degeneration-related genes, *Mmp3* and *Mmp13*. These data demonstrated that miR-455s regulate cartilage homeostasis likely by repressing Hif-2 α expression and may represent a therapeutic target for OA.

The miR-455 knockout mice showed an OA-like phenotype, which is consistent with a recent report of 5-mo miR-455-3p deletion mice²⁷. The mice in that report were generated by partial deletion of the miR-455-3p sequence, which is expected to inhibit the formation of the pre-miR-455 structure, and thus inhibit expression of both miR-455-5p and -3p. This report also indicated that miR-455-3p directly targets *PAK2*, which they showed was an inhibitor of TGF- β /Smad signaling that plays a key role in maintaining cartilage homeostasis²⁷. They also indicated that the expression of *PAK2* is upregulated in OA cartilage²⁷. *PAK2* knockdown as well as overexpression of miR-455-3p in human OA chondrocytes correlated with increased expression of cartilage-specific genes *SOX9*, *COL2A1* and *ACAN*, and decreased expression of cartilage degeneration-related genes *RUNX2*, *COL10A1* and *MMP13*²⁷. Thus, *PAK2* and HIF-2 α are thought to be critical genes for cartilage degeneration. Targeting these two genes by miR-455 might contribute more effectively to maintain cartilage homeostasis. We propose that the suppression of different genes that promote cartilage degeneration by two miRNAs generated from one pre-miRNA more effectively controls cartilage homeostasis.

Minor Comments:

1. The number of experiments should be added in the Figure Legend.

We thank you for noticing. The number of samples and experiments are now stated in all Figure legends. Changes in the revised MS are marked in red.

Reviewers' Comments:

Reviewer #1:

Remarks to the Author:

The authors have been very responsive to the concerns raised in the previous review. The additional data and revised text have strengthened the paper.

However, responses by the authors in the Reporting Summary form raise some additional questions.

For example, in the "Life sciences study design" section, the authors state that qPCR analyses were run once, using at least 3 distinct samples. The meaning of this is clear for analysis of samples from in vivo studies, as each sample is from one individual. However, for in vitro studies, the meaning is less clear. Were these experiments performed only once with 3 biological replicates or was each experiment repeated several times, with similar results?

In the Editorial Policy Checklist form, the authors attest that individual data points are shown whenever possible, and must be shown when N is less than 10. However, most of the data shown in bar graphs are mean plus or minus standard error of the mean. These data should be shown as box plots or dot plots.

Line 169/170: The authors state: ... There were no significant changes in bone formation of the femur and skull in miR-455-/- mice.

However, the authors did not evaluate bone formation, per se. They evaluated bone morphometry and microarchitecture. This statement should be corrected.

Line 230/230 The authors stateRegarding the surrounding muscles (quadriceps femoris), no difference was observed between the sham group and the miRNA-transfected group (Extended Data Fig. 12).

The authors should state exactly what assays were performed to determine that there were no differences between the groups.

Reviewer #2:

Remarks to the Author:

The statistical issue remains critical. The authors did not understand the conditions that are appropriate for ttest. Again, ttest is not the right method for analyzing data with more than 2 groups. The authors can not just or only do ttest on the selected 2 groups they wanted to do from a few groups of data.

The authors should clarify the meaning of injection of knee. Does it mean the injection into intra articular space of the knee or into the articular cartilage?

The other questions were addressed.

Reviewer #3:

Remarks to the Author:

It is commendable that the authors have done a great deal to improve the quality of the articles and have addressed most of the issues. However, the reviewer still has a few questions and hopes authors to solve them.

1.The authors mentioned Mir-455-3p and 5P have synergistic effects. and then what is the mechanism? The authors should discuss it in more detail.

2.There is some doubt about direct injection with mimic.The reviewer believes that AAV-mimic

may be better and more convincing.

3. The reviewer noticed that the authors identified several new targets of miR-455s. Why did the authors choose the Epas1? Does it mean that other targets are not as important as Epas1?

REVIEWER COMMENTS:

Reviewer #1 (Remarks to the Author):

The authors have been very responsive to the concerns raised in the previous review. The additional data and revised text have strengthened the paper.

However, responses by the authors in the Reporting Summary form raise some additional questions.

For example, in the “Life sciences study design” section, the authors state that qPCR analyses were run once, using at least 3 distinct samples. The meaning of this is clear for analysis of samples from in vivo studies, as each sample is from one individual. However, for in vitro studies, the meaning is less clear. Were these experiments performed only once with 3 biological replicates or was each experiment repeated several times, with similar results?

Thank you for your comment. The qPCR analyses such as confirmation of knockout and overexpression, the expression change of known cartilage marker genes, and confirmation of miRNA expression of array results were performed in triplicate in a single experiment. The others were performed independently at least twice, in duplicate or triplicate. We have revised all the figure legends so that this is now stated and have added the source data file to the MS.

In the Editorial Policy Checklist form, the authors attest that individual data points are shown whenever possible, and must be shown when N is less than 10. However, most of the data shown in bar graphs are mean plus or minus standard error of the mean. These data should be shown as box plots or dot plots.

Thank you for the note – we have now replaced all graphs where $N < 10$ (Fig. 1b, e, 2d, 3b, c, 4b, d, f, 5e, 6b, sFig. 1, 2b, 3, 4, 6, 7, 11, 12a, b, 14, and 15) with dot plots. In addition, we have added the source data file to the MS.

Line 169/170: The authors state: ... There were no significant changes in bone formation of the femur and skull in miR-455^{-/-} mice.

However, the authors did not evaluate bone formation, per se. They evaluated bone

morphometry and microarchitecture. This statement should be corrected.

We apologise for the error and have corrected the statement as follows:

OA-like pathology is observed in miR-455 knockout mouse knee joints

There were no significant changes in **bone morphometry and microarchitecture** of the femur and skull in miR-455^{-/-} mice.

Line 230/230 The authors stateRegarding the surrounding muscles (quadriceps femoris), no difference was observed between the sham group and the miRNA-transfected group (Extended Data Fig. 12).

The authors should state exactly what assays were performed to determine that there were no differences between the groups.

We apologise for the omission. For the muscle analysis, the knee joints of the sham, negative control mimic and miR-455-5p/3p groups were HE-stained and the structural changes in muscle fibers, fibrosis, and infiltration of inflammatory cells were observed in the quadriceps femoris. As a result, no changes were observed between the sham and miRNA mimic-transfected groups. The following text was added to the "Histological analysis" section of the method:

Histological analysis

For the muscle analysis, the knee joints of the sham (n=4), negative control mimic (n=5) and miR-455-5p/3p mimic groups (n=5) were H&E-stained and the structural changes in muscle fibers, fibrosis, and infiltration of inflammatory cells were observed in the quadriceps femoris.

Reviewer #2 (Remarks to the Author):

The statistical issue remains critical. The authors did not understand the conditions that are appropriate for ttest. Again, ttest is not the right method for analyzing data with more than 2 groups. The authors can not just or only do ttest on the selected 2 groups they wanted to do from a few groups of data.

We apologise for the lack of understanding. Statistical comparison between more than two groups were analyzed by Dunnett's tests or Tukey-Kramer tests. Dunnett's tests and Tukey-Kramer tests were performed using RStudio.Version (3.5.1).

Details of the statistical analyses are now added to all figure legends and the “Statistical analysis” section has been revised as follows:

Statistical analysis

Statistically significant differences between two groups were evaluated using the two-tailed Student's t-tests. Significant differences between more than two groups were analyzed by Dunnett's tests (one- or two-tailed) or Tukey–Kramer tests. Dunnett's tests and Tukey-Kramer tests were performed using RStudio.Version (3.5.1) (RStudio Team (2020). RStudio: Integrated Development for R. RStudio, PBC, Boston, MA URL <http://www.rstudio.com/>). Differences were considered significant at $P < 0.05$ (* = $P < 0.05$, ** = $P < 0.01$, *** = $P < 0.001$).

The authors should clarify the meaning of injection of knee. Does it mean the injection into intra articular space of the knee or into the articular cartilage?

We apologise for the confusion. We injected into the intra articular space of the knee. The “OA-model mouse” section has been revised as follows:

OA-model mice

C57BL/6J mice were used for this study. All animal procedures were performed according to protocols approved by the Institutional Animal Care and Use Committee (IACUC) at The Scripps Research Institute. All mice were freely allowed access to food, water, and activity. In the treatment study, OA was surgically induced by DMM in the right knee joints of 3-mo mice⁶⁸. The left knees were subjected to sham surgery. The target mimic conjugated hydrogel was created as described previously⁶⁴. In brief, 1.5 μ l miRNA mimic (50 μ M), 1.5 μ l completion buffer, and 3 μ l invivofermine 3.0 reagent (Thermo) were mixed, followed by incubation at 50 °C for 30 minutes. Then, 15 μ l PBS, 7.5 μ l AteloGene QG (KOKEN), and 7.5 μ l dilution buffer were added. **30 μ l of the mixture was injected into the intra articular space of the knee.** The mice (n=48) were divided into four groups. Group 1 (control group) was treated with control miRNA mimic (Pre-miR, Ambion). Group 2 was treated with mouse miR-455-3p (Ambion). Group 3 was treated with mouse miR-455-5p (Ambion). Group 4 was treated with miR-455-3p and miR-455-5p. Two days after DMM, the treatment of mimic with an injection to the right knee started 2 times a week for 10 weeks and the mice were killed at 10 weeks after surgery. The entire knee joints were fixed in 10% zinc-buffered formalin for 2 days and decalcified in TBD-2 for 24 h. To investigate whether the

mimic reached the cartilage, we created two additional groups that were treated with only mimic injection 2 times a week for 10 weeks. One was treated with control miRNA mimic, and the other with miR-455-5p and -3p (n=3, each). The efficacy of the miR-455 mimic was examined by qPCR from RNA extraction from cartilage 3 days after the final injection (Extended Data Fig. 12). To test whether the injection of the miRNA mimic affects the phenotype, we created three additional groups that were treated only with mimic injection 2 times a week for 10 weeks. Group 1 was treated with control miRNA mimic (n=6), Group 2 with miR-455-5p and -3p (n=8), and group 3 was subjected to sham surgery (n=6). The entire knee joints were fixed in 10% formalin for a day and decalcified in formic acid/sodium citrate solution for 24 h. The samples were embedded and stained with Safranin-O or hematoxylin and eosin.

The other questions were addressed.

Reviewer #3 (Remarks to the Author):

It is commendable that the authors have done a great deal to improve the quality of the articles and have addressed most of the issues. However, the reviewer still has a few questions and hopes authors to solve them.

1. The authors mentioned Mir-455-3p and 5p have synergistic effects. and then what is the mechanism? The authors should discuss it in more detail.

We appreciate your comment. Although we were unable to directly demonstrate a synergistic inhibitory effect of miR-455-5p and -3p on cartilage degeneration in the present study, we have shown that injection of either the miR-455-5p or -3p mimic failed to inhibit cartilage degeneration in the DMM-treated knee joint, whereas injection of both miR-455-5p and -3p inhibited cartilage destruction in the DMM-treated knee joint. In addition, in terms of the mechanism, *Epas1* knockdown rescued increased expression of cartilage degeneration-related genes in the knee cartilage of 6-mo miR-455 knockout mice, indicating that the phenotype of cartilage degeneration in miR-455 knockout mice is due, at least in part, to the increased expression of *Epas1* in the cartilage. Furthermore, mutation of the *Epas1* reporter at either miR-455-5p or -3p target sites abolished the inhibitory effect of miR-455 on the *Epas1* reporter. Based on these results, we speculate that the mechanism of this apparent 'additive' inhibitory effect of miR-455-5p and -3p on chondrogenic degeneration may be due, at least in part, to the coordinated repression of *Epas1* expression by miR-455-5p and -3p. We have adjusted the Discussion to reflect this better as follows.

Discussion

In the present study, we **suggest that there may be an additive effect** of miR-455-5p and -3p in the treatment of experimental OA. Both miR-455-5p and -3p coordinately regulate expression of HIF-2 α , a central transactivator of catabolic factors^{41, 42}. Although HIF-2 α , encoded by the *EPAS1* gene, is a potential therapeutic target for OA, since *EPAS1* is an important gene for development, *Epas1*^{-/-} mice are embryonic lethal, and *Epas1*^{+/-} mice show dwarfism⁴¹. Therefore, indirect suppression of *EPAS1* by miR-455s may be safer than direct siRNA knockdown of *EPAS1* as a treatment for OA. Also, injection of both miR-455-5p and -3p into intra-articular cartilage was more effective in alleviating experimental OA compared to injection of miR-455-5p or -3p alone, further supporting their **additive activity**. These data also indicate that using multiple miRNAs in OA treatment can be more functional. In this regard, we have also demonstrated the **additive effect** of a miRNA and the host mRNA in an OA model⁶⁵. These data indicate that combining multiple miRNA injections may lead to more effective treatments for OA.

2. There is some doubt about direct injection with mimic. The reviewer believes that AAV-mimic may be better and more convincing.

Thank you for your comment. Although AAV is a relatively safe viral vector and has many advantages over mimic injections in terms of persistence of expression, amount of expression and efficiency of transduction, we considered that introduction of miRNAs using a non-viral vector would be a clinically preferred method for the treatment of OA, which is not a severe life-threatening disease. Thus, we utilized a transfection method for miRNA introduction. We have revised the “Introduction of both miR-455-5p and -3p inhibits cartilage degeneration” in the Results section to better explain that as follows:

Introduction of both miR-455-5p and -3p inhibits cartilage degeneration

To investigate the potential therapeutic effect of miR-455s, we used the well-established surgical destabilization of the medial meniscus (DMM) model of OA injected with miR-455s mimics (Fig. 5a). **We used the transfection method for introduction of miRNAs, which is safer than using viral vectors and has lower hurdles for medical applications.** We first investigated whether introduction of miRNA mimics had an impact on articular cartilage and the phenotype of the surrounding tissue in the absence of DMM surgery.

3. The reviewer noticed that the authors identified several new targets of miR-455s. Why did the authors choose the *Epas1*? Does it mean that other targets are not as important as *Epas1*?

We appreciate your comment. Because *Epas1* was already known to be associated with cartilage degeneration, we focused on that. However, there is a paper that suggests that one of the other targets, *Zbtb20*, is involved in hypertrophic cartilage differentiation by suppressing *Sox9*, so it is quite possible that some of the other targets may be involved in maintaining cartilage homeostasis.

We added the following paragraph to the Discussion:

We identified *EPAS1*, *APH1A*, *FBXL2*, *FKBPL* and *ZBTB20* as novel target genes of miR-455 by using a reporter library system. Although we focused on *EPAS1* alone in this study, it is conceivable that some of the other genes may be important target genes mediating the inhibitory effect of miR-455 on cartilage degeneration. For example, *Zbtb20* has been reported to be important in the terminal differentiation of hypertrophic chondrocytes by repressing *Sox9*⁵⁵. It is possible that *Zbtb20* is involved in cartilage homeostasis as well as development by *Sox9* repression, and that miR-455 may contribute to cartilage homeostasis also by repressing *Zbtb20*.

Reviewers' Comments:

Reviewer #2:

Remarks to the Author:

The issues were addressed.

Reviewer #3:

Remarks to the Author:

I do not have any other questions about this article.